Corrected: Author correction

# Cross-ancestry genome-wide association analysis of corneal thickness strengthens link between complex and Mendelian eye diseases

Adriana I. Iglesias et al.#

Central corneal thickness (CCT) is a highly heritable trait associated with complex eye diseases such as keratoconus and glaucoma. We perform a genome-wide association meta-analysis of CCT and identify 19 novel regions. In addition to adding support for known connective tissue-related pathways, pathway analyses uncover previously unreported gene sets. Remarkably, >20% of the CCT-loci are near or within Mendelian disorder genes. These included *FBN1*, *ADAMTS2* and *TGFB2* which associate with connective tissue disorders (Marfan, Ehlers-Danlos and Loeys-Dietz syndromes), and the *LUM-DCN-KERA* gene complex involved in myopia, corneal dystrophies and cornea plana. Using index CCT-increasing variants, we find a significant inverse correlation in effect sizes between CCT and keratoconus ($r = -0.62$, $P = 5.30 \times 10^{-5}$) but not between CCT and primary open-angle glaucoma ($r = -0.17$, $P = 0.2$). Our findings provide evidence for shared genetic influences between CCT and keratoconus, and implicate candidate genes acting in collagen and extracellular matrix regulation.

#A full list of authors appears at the end of the paper.

Central corneal thickness (CCT) is a highly heritable quantitative trait, with heritability estimates ranging between 0.68 and 0.95[1–4]. The corneal stroma, which accounts for 90% of the corneal thickness in humans, is composed of uniformly arranged type I collagen fibrils that are critical to the optical properties of the cornea. Corneal thinning is a common feature of rare Mendelian connective tissue disorders, such as Ehlers-Danlos syndrome (EDS), Marfan syndrome and osteogenesis imperfecta (OI), and extreme thinning is a clinical characteristic of brittle cornea syndrome (previously classified as EDS type VIB based on non-ocular features shared with EDS)[5–7]. Thinner CCT is also observed in more common ocular disorders such as keratoconus[8], and has been associated with development and progression of primary open-angle glaucoma (POAG)[9–12]. Keratoconus is the leading cause of corneal transplants worldwide[13] and its prevalence varies widely depending on the ethnicity, ranging from 0.02 in 100,000 to 229 in 100,000[14,15]. POAG accounts for around 74% of all cases of glaucoma, which is the most common cause of irreversible blindness worldwide[16].

Genetic variants that affect the functions of genes responsible for maintaining the structural integrity of cornea are strong candidates for involvement in corneal thickness-associated diseases. We previously reported that six CCT-associated single nucleotide polymorphisms (SNPs) were also associated with keratoconus (using $N = 874$ cases and 6085 controls), including two strong associations (mean odds ratio and lower 95% confidence interval estimates greater than 1.2), at the FOXO1 and FNDC3B loci[17]. The latter was also shown to be associated with POAG (using $N = 2979$ cases and 7399 controls), although not in the direction that was expected (i.e., the CCT-decreasing allele was associated with decreased risk of POAG)[17].

Over 26 loci have been associated with CCT to date, explaining around 8% of the CCT heritability[17]. Increased knowledge of the genetic basis of the variation in CCT in the general population promises to help in prioritising future research in corneal disease. To identify new CCT-associated loci, we performed a larger cross-ancestry genome-wide association study (GWAS) including over 25,000 individuals of European and Asian descent, with genotypes imputed to the 1000 genomes reference panel. Further, we assess the relevance of CCT influencing loci to the risk of

keratoconus and POAG using slightly larger (keratoconus) and substantially larger (POAG) ocular disease datasets than those previously described[17].

## Results

**Meta-analysis of GWAS studies.** The overall study design and main findings are depicted in Supplementary Fig. 1. In stage 1, we meta-analysed GWAS results from 14 studies comprising 17,803 individuals of European ancestry (see details in Supplementary Table 1). The inflation factor for European-specific meta-analysis was 1.075 (lambda scaled to $n = 1000$ is 1.004), which suggests the population stratification had a negligible effect on our meta-analysis. The European-specific meta-analysis identified 28 genome-wide significant CCT loci ($P < 5 \times 10^{-8}$) (Supplementary Table 2 and Supplementary Fig. 2a, b). Of these, seven were novel loci and map (as per closest gene) to LTBP1, STAG1, ARL4C, NDUFAF6, ADAMTS8, DCN and POLR2A. In stage 2, we examined the 28 lead SNPs from stage 1 in the Asian-specific meta-analysis ($n = 8107$) and found that 16, including the novel lead SNPs within or close to ADAMTS8 and DCN, were significant after Bonferroni correction ($P \leq 1.79 \times 10^{-3}$, 0.05/28), further three other SNPs including the two novel close to STAG1 and NDUFAF6 were nominally significant ($P < 0.05$). The effect estimates of these 19 (16 + 3) loci were in the same direction and order of magnitude as in the European-specific meta-analysis (Tables 1 and 2 and Supplementary Table 2). Lead SNPs at four of the nine remaining loci, including at LTBP1, did not meet our filtering criteria in the Asian-specific meta-analysis (see Methods section). Lead SNPs at the remaining five loci showed the same direction but did not reach nominal significance, with SNPs at ARL4C and POLR2A displaying little effect in Asian populations. Meta-analysis of Asian-specific cohorts alone did not result in novel genome-wide significant findings (Supplementary Table 3 and Supplementary Fig. 3a, b). Because most loci had consistent effect directions in both European and Asian meta-analyses, we performed in stage 3 a cross-ancestry fixed-effect meta-analysis to detect additional loci associated with CCT ($N = 25,910$). This stage 3 meta-analysis identified 44 loci associated with CCT of which 19 were novel findings (Fig. 1 and Supplementary

---

### Table 1 Results from cross-ancestry meta-analysis (chromosomes 1–7)

| SNP | Chr:bp | Nearest gene | A1/A2 | European-specific meta-analysis | | | Asian-specific meta-analysis | | | Cross-ancestry meta-analysis | | | |
|---|---|---|---|---|---|---|---|---|---|---|---|---|---|
| | | | | A1F | β (SE) | P | A1F | β (SE) | P | A1F | β (SE) | P | N |
| rs96067 | 1:36571920 | COL8A2 | a/g | 0.81 | 0.99 (0.48) | 4.08E-02 | 0.56 | 3.94 (0.52) | 3.48E-14 | 0.69 | 2.37 (0.35) | 2.52E-11 | 25,910 |
| **rs4846476** | **1:218526228** | **TGFB2** | **c/g** | **0.23** | **−1.83 (0.46)** | **7.22E-05** | **0.31** | **−2.11 (0.56)** | **1.64E-04** | **0.26** | **−1.94 (0.35)** | **4.77E-08** | **23,830** |
| **rs115781177** | **2:33348494** | **LTBP1** | **a/g** | **0.93** | **−5.04 (0.89)** | **1.69E-08** | **NA** | **NA (NA)** | **NA** | **0.93** | **−5.04 (0.89)** | **1.69E-08** | **12,119** |
| rs121908120 | 2:219755011 | WNT10A | a/t | 0.03 | −11.48 (1.58) | 5.02E-13 | NA | NA (NA) | NA | 0.03 | −11.48 (1.58) | 5.02E-13 | 12,119 |
| rs4608502 | 2:228134155 | COL4A3 | t/c | 0.35 | −2.47 (0.39) | 4.68E-10 | 0.36 | −2.19 (0.54) | 5.12E-05 | 0.35 | −2.37 (0.32) | 1.18E-13 | 25,910 |
| **3:136138073** | **3:136138073** | **STAG1** | **d/r** | **0.24** | **−2.64 (0.47)** | **2.49E-08** | **0.19** | **−2.80 (1.09)** | **1.05E-02** | **0.23** | **−2.67 (0.43)** | **8.66E-10** | **20,982** |
| rs9822953 | 3:156472071 | TIPARP[a] | t/c | 0.67 | 2.69 (0.40) | 2.57E-11 | 0.67 | 1.11 (0.61) | 7.15E-02 | 0.67 | 2.22 (0.33) | 5.13E-11 | 25,910 |
| rs6445046 | 3:171933252 | FNDC3B | t/g | 0.78 | 3.73 (0.49) | 7.22E-14 | 0.66 | 3.17 (0.57) | 3.59E-08 | 0.73 | 3.49 (0.37) | 1.98E-20 | 24,899 |
| 3:177306757 | 3:177306757 | TBL1XR1[b] | d/r | 0.39 | −2.40 (0.42) | 1.43E-08 | 0.53 | −1.84 (0.52) | 4.35E-04 | 0.44 | −2.18 (0.32) | 3.54E-11 | 23,060 |
| rs28789690 | 4:149077899 | NR3C2 | a/g | 0.07 | −3.02 (0.74) | 4.93E-05 | 0.11 | −3.49 (0.84) | 3.59E-05 | 0.09 | −3.22 (0.55) | 7.60E-09 | 25,128 |
| rs10471310 | 5:64548961 | ADAMTS6 | t/c | 0.37 | 2.62 (0.39) | 1.74E-11 | 0.39 | 1.87 (0.53) | 4.36E-04 | 0.38 | 2.36 (0.31) | 6.12E-14 | 25,910 |
| **rs249767** | **5:141918585** | **FGF1** | **t/c** | **0.78** | **2.01 (0.46)** | **1.56E-05** | **0.51** | **2.15 (0.54)** | **7.30E-05** | **0.67** | **2.07 (0.35)** | **4.60E-09** | **25,910** |
| **rs35028368** | **5:178671146** | **ADAMTS2** | **i/r** | **0.29** | **−2.34 (0.48)** | **1.25E-06** | **0.11** | **−2.59 (0.99)** | **8.80E-03** | **0.26** | **−2.39 (0.43)** | **3.69E-08** | **23,060** |
| **rs13191376** | **6:45522139** | **RUNX2** | **t/c** | **0.35** | **−2.07 (0.39)** | **1.78E-07** | **0.14** | **−1.99 (0.91)** | **2.94E-02** | **0.32** | **−2.06 (0.36)** | **1.55E-08** | **25,910** |
| **rs1412710** | **6:75837203** | **COL12A1** | **t/c** | **0.15** | **−2.56 (0.56)** | **5.26E-06** | **0.33** | **−1.93 (0.58)** | **9.19E-04** | **0.24** | **−2.26 (0.40)** | **2.42E-08** | **24,899** |
| rs1931656 | 6:82610188 | FAM46A | a/t | 0.45 | 2.17 (0.39) | 2.75E-08 | 0.47 | 2.96 (0.52) | 2.13E-08 | 0.46 | 2.451 (0.32) | 6.32E-15 | 24,899 |
| **6:169553553** | **6:169553553** | **THBS2** | **i/r** | **0.19** | **−2.98 (0.62)** | **1.76E-06** | **0.30** | **−2.22 (0.69)** | **1.41E-03** | **0.24** | **−2.64 (0.46)** | **1.27E-08** | **23,060** |
| 7:66262284 | 7:66262284 | RABGEF1[c] | d/r | 0.27 | −3.32 (0.44) | 1.25E-13 | 0.34 | −2.73 (0.56) | 1.03E-06 | 0.29 | −3.09 (0.35) | 9.62E-19 | 24,071 |
| **rs2106166** | **7:92668332** | **SAMD9** | **a/t** | **0.57** | **1.95 (0.40)** | **1.39E-06** | **0.38** | **1.48 (0.55)** | **7.99E-03** | **0.50** | **1.79 (0.32)** | **4.63E-08** | **24,899** |

Nearest gene (reference NCBI build37) is given as locus label, but this should not be interpreted as providing support that the nearest gene is the best candidate, a list including all the genes +/− 200 kb of the lead SNP is presented in Supplementary Table 12
New loci are in bold
SNP rsID, Chr:bp chromosome: base pair, A1 risk allele, A2 other allele, A1F frequency of allele A1, β effect size on CCT based on allele A1, SE standard error of the effect size, i insertion, d deletion, r reference, N number of individuals included in the meta-analysis per variant
[a]The lead SNP is located in a validated non-coding mRNA, LINC00886
[b]The lead SNP is located in a validated non-coding mRNA, LINC00578
[c]In Lu et al.[17] this locus was reported as two loci (VKORC1L1 and C7orf42)

**Table 2 Results from cross-ancestry meta-analysis (chromosomes 8–22)**

| SNP | Chr:bp | Nearest gene | A1/A2 | European-specific meta-analysis | | | Asian-specific meta-analysis | | | Cross-ancestry meta-analysis | | | |
|---|---|---|---|---|---|---|---|---|---|---|---|---|---|
| | | | | A1F | β (SE) | P | A1F | β (SE) | P | A1F | β (SE) | P | N |
| **rs3808520** | **8:23164773** | *LOXL2* | **c/g** | **0.21** | **2.50 (0.48)** | **2.61E-07** | **0.10** | **2.02 (0.88)** | **2.28E-02** | **0.18** | **2.39 (0.42)** | **2.02E-08** | **24,899** |
| **rs10429294** | **8:95969322** | *NDUFAF6* | **t/c** | **0.50** | **2.36 (0.39)** | **2.21E-09** | **0.66** | **1.34 (0.55)** | **1.60E-02** | **0.56** | **2.02 (0.32)** | **3.48E-10** | **24,899** |
| **rs7026684** | **9:4215308** | *GLIS3* | **a/g** | **0.36** | **−2.00 (0.39)** | **4.24E-07** | **0.39** | **−1.81 (0.55)** | **1.04E-03** | **0.37** | **−1.94 (0.32)** | **1.73E-09** | **25,910** |
| rs66720556 | 9:13559717 | *MPDZ* | a/t | 0.18 | −1.86 (0.51) | 3.01E-04 | 0.25 | −3.80 (0.59) | 2.18E-10 | 0.21 | −2.68 (0.39) | 6.06E-12 | 24,071 |
| rs10980623 | 9:113660537 | *LPAR1* | a/g | 0.79 | −2.63 (0.46) | 1.06E-08 | 0.79 | −3.43 (0.63) | 6.06E-08 | 0.79 | −2.90 (0.37) | 5.63E-15 | 25,910 |
| rs3132303 | 9:137444298 | *COL5A1* | c/g | 0.26 | 5.23 (0.49) | 6.11E-26 | 0.26 | 5.91 (0.71) | 1.21E-16 | 0.26 | 5.45 (0.40) | 8.35E-41 | 24,899 |
| rs7040970 | 9:139859013 | *LCN12* | t/c | 0.49 | 3.35 (0.41) | 3.54E-16 | 0.72 | 1.80 (0.63) | 4.31E-03 | 0.56 | 2.89 (0.34) | 4.75E-17 | 24,899 |
| rs35809595 | 10:63831928 | *ARID5B* | a/g | 0.40 | −2.29 (0.39) | 8.97E-09 | 0.36 | −2.66 (0.53) | 6.56E-07 | 0.39 | −2.43 (0.32) | 3.40E-14 | 24,899 |
| **rs2419835** | **10:115296564** | *HABP2* | **t/c** | **0.86** | **2.21 (0.54)** | **4.70E-05** | **0.45** | **2.33 (0.52)** | **9.01E-06** | **0.65** | **2.27 (0.37)** | **1.74E-09** | **25,910** |
| rs4938174 | 11:110913240 | *ARHGAP20-C11orf53* | a/g | 0.30 | 1.82 (0.41) | 9.97E-06 | 0.15 | 3.74 (0.75) | 6.14E-07 | 0.26 | 2.26 (0.36) | 3.59E-10 | 25,910 |
| **rs56009602** | **11:130289612** | *ADAMTS8* | **t/c** | **0.05** | **6.86 (0.92)** | **1.30E-13** | **0.10** | **7.24 (0.93)** | **1.25E-14** | **0.08** | **7.05 (0.66)** | **1.16E-26** | **25,910** |
| **rs7308752** | **12:91527181** | *DCN* | **a/g** | **0.91** | **3.87 (0.67)** | **1.07E-08** | **0.73** | **2.28 (0.68)** | **7.91E-04** | **0.82** | **3.08 (0.48)** | **1.34E-10** | **25,302** |
| rs11553764 | 12:104415244 | *GLT8D2* | t/c | 0.17 | 3.19 (0.53) | 2.77E-09 | 0.20 | 4.14 (0.67) | 8.62E-10 | 0.18 | 3.55 (0.42) | 2.47E-17 | 24,899 |
| rs10161679 | 13:23243645 | *FGF9-SGCG*[a] | a/g | 0.71 | −2.40 (0.45) | 1.41E-07 | 0.72 | −1.99 (0.64) | 2.16E-03 | 0.71 | −2.26 (0.37) | 1.28E-09 | 24,899 |
| 13:41112152 | 13:41112152 | *FOXO1* | i/r | 0.10 | −5.44 (0.66) | 2.15E-16 | 0.03 | −2.52 (1.81) | 1.64E-01 | 0.10 | −5.10 (0.62) | 2.54E-16 | 24,071 |
| **rs56223983** | **14:81814754** | *STON2* | **t/g** | **0.30** | **2.01 (0.42)** | **1.83E-06** | **0.30** | **1.99 (0.58)** | **5.94E-04** | **0.30** | **2.00 (0.34)** | **4.14E-09** | **25,910** |
| rs785422 | 15:30173885 | *TJP1* | t/c | 0.11 | −4.01 (0.63) | 2.65E-10 | 0.08 | −3.91 (1.26) | 5.75E-03 | 0.10 | −3.91 (0.56) | 5.72E-12 | 21,810 |
| **rs8030753** | **15:48801935** | *FBN1* | **t/c** | **0.13** | **2.02 (0.55)** | **2.75E-04** | **0.27** | **2.51 (0.59)** | **2.29E-05** | **0.20** | **2.25 (0.40)** | **2.87E-08** | **25,910** |
| rs12912010 | 15:67467143 | *SMAD3* | t/g | 0.22 | 2.76 (0.47) | 6.40E-09 | 0.36 | 2.21 (0.53) | 3.92E-05 | 0.28 | 2.52 (0.35) | 1.50E-12 | 24,899 |
| rs4843040 | 15:85838636 | *AKAP13*[b] | t/c | 0.24 | −2.92 (0.44) | 3.62E-11 | 0.47 | −2.35 (0.52) | 6.68E-06 | 0.33 | −2.68 (0.33) | 1.68E-15 | 25,910 |
| rs930847 | 15:101558562 | *LRRK1* | t/c | 0.77 | −3.57 (0.45) | 3.19E-15 | 0.73 | −3.79 (0.61) | 7.82E-10 | 0.76 | −3.64 (0.36) | 1.63E-23 | 25,910 |
| rs35193497 | 16:88324821 | *ZNF469* | t/g | 0.36 | −6.23 (0.43) | 8.64E-47 | 0.29 | −4.92 (0.62) | 2.34E-15 | 0.34 | −5.80 (0.35) | 8.08E-60 | 24,899 |
| rs4792535 | 17:14565130 | *HS3ST3B1* | t/c | 0.29 | −2.43 (0.41) | 3.61E-09 | 0.47 | −2.04 (0.54) | 1.72E-04 | 0.36 | −2.29 (0.32) | 3.13E-12 | 25,302 |
| **rs8133436** | **21:47519535** | *COL6A2* | **t/c** | **0.05** | **3.90 (1.07)** | **2.84E-04** | **0.25** | **3.47 (0.72)** | **1.85E-06** | **0.18** | **3.61 (0.60)** | **2.17E-09** | **24,899** |
| **rs71313931** | **22:19960184** | *ARVCF* | **c/g** | **0.71** | **−2.23 (0.44)** | **5.49E-07** | **0.78** | **−2.22 (0.70)** | **1.59E-03** | **0.73** | **−2.23 (0.37)** | **3.21E-09** | **24,071** |

Nearest gene (reference NCBI build37) is given as locus label, but this should not be interpreted as providing support that the nearest gene is the best candidate, a list including all the genes +/− 200 kb of the lead SNP is presented in Supplementary Table 12
New loci are in bold
*SNP* rsID, *Chr:bp* chromosome: base pair, *A1* risk allele, *A2* other allele, *A1F* frequency of allele A1, *β* effect size on CCT based on allele A1, *SE* standard error of the effect size, *i* insertion, *d* deletion, *r* reference, *N* number of individuals included in the meta-analysis per variant
[a]The lead SNP is located 228KB 3' of the pseudogene *BASP1P1*
[b]The lead SNP is located in pseudogene *ADAMTS7P4*

Figs. 4, 5). These 19 loci included five of the seven loci found in stage 1 (European-meta-analysis) and 14 novel ones with similar effect size and direction across-ancestries, see Tables 1 and 2. Two of the 44 loci are driven by low-frequency variants (i.e., 0.01 < minor allele frequency [MAF] < 0.05) identified in the European-specific meta-analysis (both are monomorphic in Asians), one novel in *LTBP1* and one known in *WNT10A*[18]. The remaining 42 loci were all consistent across ancestries.

**Independent signals.** In our previous CCT GWAS we identified loci harbouring multiple independent variants[17]. To identify additional independently associated variants in European population, we performed conditional and joint multiple-SNP (CoJo) analysis implemented in the program GCTA. We used genotype data of 2582 unrelated Australians from the BMES cohort[19]. The CoJo analysis resulted in 16 independent SNPs, of which seven have not been previously associated with CCT (Table 3). Thus, in total, we identified 44 loci associated with CCT, harbouring 54 independent association signals (i.e., 28 previously published + 19 from the cross-ancestry meta-analysis + 7 from CoJo analyses).

**Gene-based association analysis.** To further identify loci not implicated in the single marker association tests, we performed gene-based tests using the software VEGAS2[20] using a '−10 kb' window (see Methods section). We performed separate analyses for European-specific and Asian-specific meta-analyses results. In total 24,769 autosomal genes were analysed. Hence, we set our Bonferroni-corrected gene-based significant threshold as $P_{\text{gene-based}} < 2.02 \times 10^{-6}$ (0.05/24,769). In addition to genes implicated through the single marker association tests, we found significant association of the *CDO1* gene with CCT ($P_{\text{gene-based}} = 3.74 \times 10^{-7}$, Supplementary Data 1). This gene showed strong association in the European gene-based study ($P_{\text{gene-based}} = 2.00 \times 10^{-7}$), with a top variant rs34869 ($P = 7.88 \times 10^{-8}$) driving the association.

**Clinical relevance of CCT-associated loci.** We first investigated whether the CCT-associated variants influence susceptibility to keratoconus and to POAG. Since keratoconus is characterized by progressive thinning of the cornea and reduced CCT is associated with POAG, we expected—if the underlying mechanisms are shared—that the CCT-reducing alleles would also increase the risks of keratoconus and of POAG. We aimed to test the association of all 54 independent CCT SNPs (or their proxies, $r^2 > 0.8$) in the case-control studies. However, after quality control, only 36 SNPs were available in the keratoconus studies; all 54 SNPs were available in the POAG studies. We used a *P*-value of $5.56 \times 10^{-4}$ (0.05/90) as Bonferroni-corrected significance threshold.

The keratoconus cohorts comprised 933 cases and 5946 controls of European ancestry. Overall, we found a significant negative correlation of effect sizes across CCT and keratoconus ($r = -0.62$, $P = 5.30 \times 10^{-5}$) (Fig. 2a, Supplementary Table 4), this correlation was largely unchanged if the known SNPs in *ZNF469, FOXO1, COL5A1* and *MPDZ/NFIB* were removed from the analysis ($r = -0.61$, $P = 2.04 \times 10^{-4}$). Of the 36 CCT SNPs tested for association with disease risk in the keratoconus studies, three were significant and with the expected direction of effect (rs66720556 between *MPDZ-NFIB*, rs3132303 between *RXRA-COL5A1*, and rs2755238 close to *FOXO1*). Another 14 independent SNPs were associated at a nominal level of significance ($P < 0.05$). Of these, 12 showed the expected risk effect direction, including three tagging known CCT loci that had not reached nominal significance in our previous study (using a different proxy SNP), and four tagging novel CCT loci (*DCN, LTBP1, STAG1*, and *THBS2*). Of those, rs7308752 in *DCN* displayed the smallest *P*-value ($P = 6.33 \times 10^{-3}$, Supplementary Table 4).

Analyses in POAG cohorts included 5008 cases and 35,472 controls of European ancestry. None of the 54 available CCT-SNPs were significantly associated with POAG after correcting for multiple testing (Supplementary Table 5). Further, no correlation in effect sizes between CCT and POAG was found ($r = -0.17$, $P = 0.2$, Fig. 2b). Five variants were nominally associated (rs6445046 in *FNDC3B*, rs66720556 between *MPDZ* and *NFIB*,

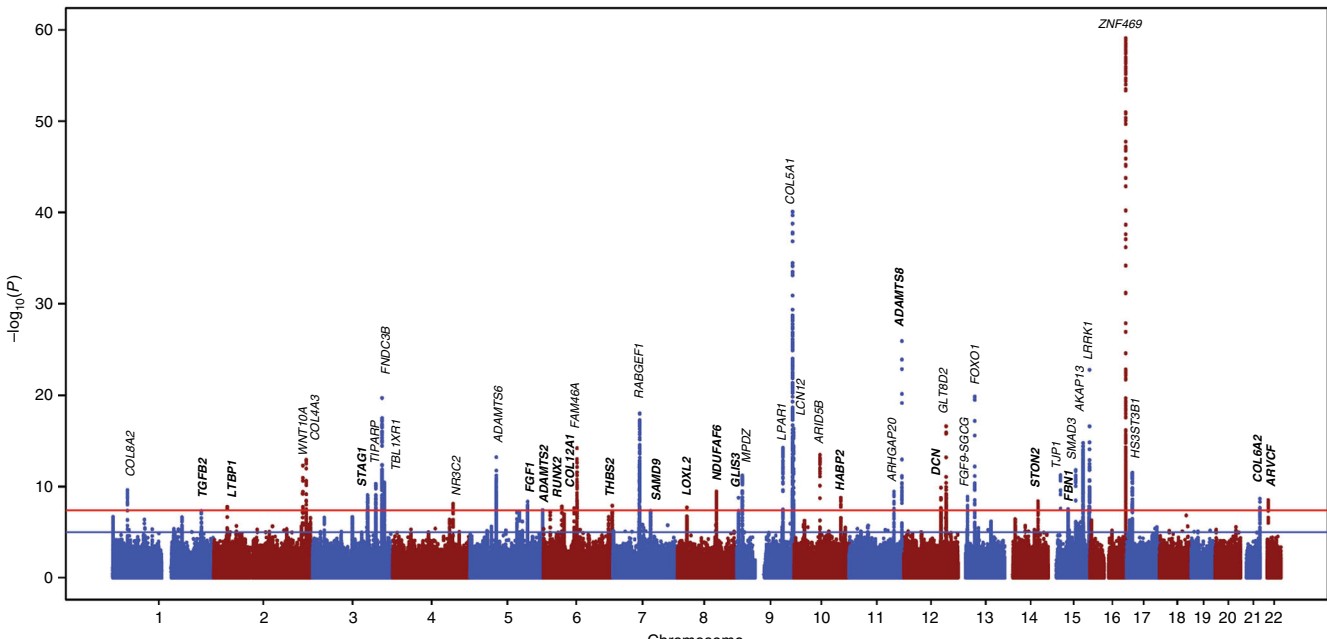

**Fig. 1** Manhattan plot of CCT in the cross-ancestry meta-analysis. Manhattan plot of the GWAS meta-analysis for CCT in the cross-ancestry analysis ($n = 25,910$). The plot shows $-\log10$-transformed $P$-values for all SNPs. The red horizontal line represents the genome-wide significance threshold of $P < 5.0 \times 10^{-8}$; the blue horizontal line indicates a $P$-value of $1 \times 10^{-5}$. Loci are annotated to the nearest gene as in Tables 1 and 2. New loci are in bold

rs56009602 in *ADAMTS8*, rs10161679 between *SNORD36* and *BASP1P1*, and rs2755238 close to *FOXO1*). Of these, association between rs6445046 in *FNDC3B* and POAG was the strongest ($P = 3.18 \times 10^{-3}$). However, as in our previous study[17], the CCT-decreasing allele tagging *FNDC3B* was associated with a decreased risk of glaucoma rather than the expected increased risk. The next strongest association was for rs2755237 close to *FOXO1* ($P = 3.70 \times 10^{-3}$); in this instance with the CCT-decreasing allele C being associated with an increased risk of glaucoma (rs2755237-C allele glaucoma OR = 1.15).

We then investigated whether CCT-associated loci are located in the vicinity (less than 1 Mb away) of rare Mendelian disorder genes (Supplementary Data 2). We identified that 20.5% (9/44) of the CCT loci are within 1 Mb of a Mendelian gene implicated in rare corneal or connective tissue diseases. In addition to the more immediate connections previously recognised (*COL5A1*—classical EDS, *ZNF469*—brittle cornea syndrome, *COL8A2*—Fuchs endothelial dystrophy), *AGBL1*—a Fuchs endothelial dystrophy gene[21]—is 784 kb away from rs4843040 on chr15q25.3, and *SMAD3* is a Loeys-Dietz connective tissue syndrome gene. For the CCT loci identified here, *DCN* (12 kb from rs7308752) and *KERA* in the same locus (75 kb from rs7308752), are involved in congenital stromal corneal dystrophy (OMIM: 610048) and Cornea plana 2 (OMIM:217300), respectively. Finally, three connective tissues disease genes harboured lead SNPs for new CCT loci, *ADAMTS2* (intronic lead SNP rs35028368) involved in EDS, type VIIC (OMIM: 225410), *FBN1* (intronic lead SNP rs8030753) the Marfan syndrome major gene (OMIM: 154700), and *TGFB2* (intronic lead SNP rs4846476) a Loeys-Dietz syndrome gene.

**Regulatory potential of CCT-associated variants**. We explored regulatory annotations within the 54 independent CCT lead SNPs and their proxies ($r^2 > 0.8$) using different tools (see Methods section). In total, 974 variants (i.e., 54 lead SNPs + 920 SNPs in LD) were examined. Of these, 118 were prioritized including the

54 lead SNPs and another 64 SNPs which were selected based on their RegulomeDB score[22] (i.e.,1a–1f, 2a–2c or 3a). SNPs with a score from 1a–1f to 2a–2c were classified as showing maximum evidence for being located in regulatory regions, while SNPs with a score of 3a were classified as showing medium evidence (Supplementary Data 3). In total, 63% (75/118) of the prioritized SNPs overlap with at least two regulatory elements of the ENCODE data (i.e., promoter or enhancer histone marks, DNase I hypersensitive sites, transcription factor or other protein-binding sites and eQTLs). Strong enrichment for histone modifications, particularly, H4K20me1 (which indicates transcriptional activation), was also found when results from the European-specific meta-analysis were assessed using GARFIELD (http://www.ebi.ac.uk/birney-srv/GARFIELD) (Supplementary Fig. 6). Additionally, we found 26 SNPs in eight loci showing a cis-eQTL effect in skin, which share the same embryonic origin as the cornea (Supplementary Data 3). Further, we tested if genes in associated CCT loci were highly expressed in any of the 209 Medical Subject Heading (MeSH) annotations used in DEPICT[23]. Tissue-enrichment analyses showed 33 FDR-associated (<0.05) tissues or cell type annotations. Of these, one annotation included the musculoskeletal system, five included tissues such as the muscle and connective tissue, and nine included cell types such as myocytes, osteoblast, chondrocytes, mesenchymal stem cells, stromal cells and fibroblasts (Supplementary Table 6). DEPICT prioritized 54 genes, of which 85% (46/54) are expressed in the human cornea. High expression levels (>200 PLIER) were observed for *SMAD3, COL12A1* and *DCN, LUM, KERA* (Supplementary Table 7), the latter three being at the same "DCN" locus.

**Pathway analysis**. We tested enrichment of the genes defined by VEGAS2 in 9981 pathways or gene-sets derived from the Biosystem's database. Using a 10 kb window in the VEGAS2 computation, we identified 23 pathways that were significantly enriched after correcting for multiple testing ($P_{\text{gene-set}} < 5.01 \times 10^{-6}$),

**Table 3 CCT-associated variants from the conditional and joint analysis of the meta-analysis of European studies and replication in Asians**

| SNP | Chr:bp | Nearest gene | Annotation | Previously reported SNP (ref)[a] | A1/A2 | Meta-analysis in Europeans | | | CoJo analysis in Europeans | | | | Meta-analysis in Asians | | |
|---|---|---|---|---|---|---|---|---|---|---|---|---|---|---|---|
| | | | | | | A1F | β (SE) | P | A1F | β (SE) | P_COJO | LD_r | A1F | β (SE) | P |
| **rs1309531** | 5:64306311 | *CWC27* | **Intronic** | | a/t | 0.55 | −2.4 (0.379) | **2.439E-10** | 0.56 | −2.096 (0.383) | 4.28E-08 | 0.130 | 0.63 | −1.184 (0.547) | 3.03E-02 |
| rs10064391 | 5:64686659 | *ADAMTS6* | Intronic | rs2307121[17] | a/g | 0.63 | −2.765 (0.397) | 3.182E-12 | 0.62 | −2.484 (0.4) | 5.53E-10 | 0.000 | 0.70 | −0.889 (0.601) | 1.39E-01 |
| **rs1931656** | 6:82610188 | *148 kb 5' of FAM46A* | **Intronic** | | a/t | 0.45 | 2.172 (0.391) | **2.749E-08** | 0.45 | 2.383 (0.393) | 1.31E-09 | −0.104 | 0.47 | 2.965 (0.529) | 2.13E-08 |
| rs9361886 | 6:82778502 | *101 kb 3' of IBTK* | Intergenic | rs1538138[17] | t/c | 0.54 | 2.391 (0.445) | 7.665E-08 | 0.56 | 2.637 (0.447) | 3.66E-09 | 0.000 | 0.57 | 2.35 (0.66) | 3.67E-04 |
| **rs3094339** | 9:136884738 | *VAV2-BRD3* | **Intergenic** | | a/g | 0.71 | −2.804 (0.426) | **4.682E-11** | 0.72 | −3.042 (0.427) | 1.01E-12 | −0.008 | 0.53 | 0.671 (0.559) | 2.30E-01 |
| rs4841899 | 9:137424412 | *92 kb 3' of RXRA* | Intergenic | rs4842044, rs1536478[66,67] | t/c | 0.67 | −2.993 (0.405) | 1.413E-13 | 0.67 | −2.289 (0.416) | 3.60E-08 | −0.037 | 0.63 | −2.383 (0.596) | 6.32E-05 |
| rs1536482 | 9:137440528 | *93 kb 5' of COL5A1* | Intergenic | rs3132306, rs3118516, rs3118520[17,68] | g/a | 0.66 | 4.569 (0.399) | 1.95E-30 | 0.66 | 3.455 (0.425) | 4.60E-16 | 0.388 | 0.68 | 2.864 (0.601) | 1.85E-06 |
| **rs3132303** | 9:137444298 | *89 kb 5' of COL5A1* | **Intergenic** | | c/g | 0.26 | 5.236 (0.497) | **6.11E-26** | 0.26 | 3.55 (0.544) | 6.86E-11 | −0.039 | 0.26 | 5.912 (0.714) | 1.21E-16 |
| rs7032489 | 9:137559775 | *COL5A1* | Intronic | rs7044529[17] | c/g | 0.86 | 4.033 (0.547) | 1.637E-13 | 0.86 | 4.296 (0.548) | 4.64E-15 | −0.008 | 0.81 | 1.845 (0.685) | 7.08E-03 |
| **rs116878472** | 12:104210992 | *NTS5DC3* | **Intronic** | | t/c | 0.97 | −8.392 (1.506) | **2.523E-08** | 0.97 | −8.829 (1.509) | 4.95E-09 | −0.058 | NA | NA | NA |
| rs11111869 | 12:104402485 | *GLT8D2* | Intronic | rs1564892[17] | g/a | 0.83 | −3.174 (0.51) | 4.77E-10 | 0.83 | −3.308 (0.511) | 9.40E-11 | 0.000 | 0.78 | −3.479 (0.636) | 4.38E-08 |
| rs2034809 | 15:101555399 | *LRRK1* | Intronic | rs4965359 | g/a | 0.51 | 1.844 (0.4) | 4.047E-06 | 0.51 | 2.545 (0.407) | 3.82E-10 | −0.177 | 0.34 | 2.161 (0.579) | 1.88E-04 |
| rs930847 | 15:101558562 | *LRRK1* | Intronic | rs930847[17] | g/t | 0.23 | 3.573 (0.453) | 3.194E-15 | 0.22 | 3.955 (0.461) | 9.17E-18 | −0.042 | 0.27 | 3.793 (0.617) | 7.82E-10 |
| **rs752092** | 15:101781934 | *CHSY1* | **Intronic** | | a/g | 0.66 | −2.205 (0.396) | **2.554E-08** | 0.67 | −2.19 (0.397) | 3.46E-08 | 0.000 | 0.79 | −1.745 (0.652) | 7.40E-03 |
| rs35193497 | 16:88324821 | *169 kb 5' of ZNF469* | Intergenic | rs6540223[17] | t/g | 0.36 | −6.238 (0.434) | 8.637E-47 | 0.34 | −4.654 (0.495) | 4.90E-21 | 0.653 | 0.29 | −4.928 (0.622) | 2.34E-15 |
| **rs28687756** | 16:88328928 | *165 kb 5' of ZNF469* | **Intergenic** | | t/g | 0.57 | −7.507 (0.584) | **8.418E-38** | 0.53 | −4.566 (0.667) | 7.84E-12 | 0.000 | NA | NA | NA |

Results from the conditional and joint analysis, genotype data from BMES cohort was used (N = 2582)

Nearest gene, (reference NCBI build37) is given as locus label, but this should not be interpreted as providing support that the nearest gene is the best candidate, a list including all the genes +/− 200 kb of the lead SNP is presented in Supplementary Table 12

*SNP* rsID, *Chr:bp* chromosome: base pair, *A1* risk allele, *A2* other allele, *A1F* frequency of allele A1, *β* effect size on CCT based on allele A1, *SE* standard error of the effect size, *i* insertion, *d* deletion, *r* reference, $P_{COJO}$ = *P*-value after CoJo analyses. In bold novel independent CCT-associated SNPs

[a]SNPs in LD ($r^2 > 0.5$) with SNP from CoJo analyses. In bold novel independent CCT-associated SNPs

Supplementary Data 4. The majority of these gene-sets are involved in the metabolic activities associated with collagen and extracellular matrix (ECM). We confirmed the previously identified significant association of the collagen trimer pathway (GO:0005581) with CCT[17]. Additional pathways involved in basement membrane (GO:0005604), TGF-β regulation (GO:0071636) and skeletal system development (GO:0001501), were also identified as associated with CCT. Similar pathways were observed using single variants with sub-threshold association *P*-values < $1 \times 10^{-4}$ as input for the interval enrichment analysis (INRICH) method, Supplementary Table 8. Pathway analysis using a 200 kb window in VEGAS2, showed comparable pathways and additionally revealed the endoplasmic-reticulum-associated protein degradation (ERAD) pathway (GO:0036503) (Supplementary Data 5). The ERAD pathway also emerged as an overrepresented canonical pathway in the Ingenuity Pathway Analysis (IPA) (Supplementary Fig. 7), along with pathways related to connective tissue disorders and metabolism (Supplementary Tables 9, 10).

The FUMA platform (https://ctg.cncr.nl/software/fuma_gwas) highlighted that eight of the closest genes to the 44 CCT cross ethnic meta-analysis lead variants are amongst the 64 fibroblastic signature genes overexpressed in cancer cells that have undergone epithelial to mesenchymal transition[24]: *THBS2, COL5A1, FBN1, LOXL2, DCN, LUM, COL6A2* and *GLT8D2*.

## Discussion

In this study, we identified 44 loci associated with CCT (42 across ancestry and two European specific—*LTBP1* and *WNT10A*), 19 of which are novel findings. We also found that six of the 44 loci harbour multiple independent signals associated with CCT. Furthermore, we explored the relevance of CCT to complex eye diseases (i.e., keratoconus and POAG) and to Mendelian disorders. We found evidence of a strong inverse correlation in effect sizes between CCT and keratoconus, but not between CCT and POAG. Interestingly, 20.5% (9/44) of the CCT-associated loci are located close or within genes implicated in rare corneal or connective tissue disorders.

We confirmed all loci, except one (rs3749260 in *GPR15*), reported in our previous study by Lu et al.[17] The variant in *GPR15* found by Lu et al.[17] in the European-specific analysis did not reach genome-wide significance in either our European-specific ($P = 2.15 \times 10^{-6}$) or cross-ancestry meta-analysis ($P = 6.93 \times 10^{-5}$); this could be due to the additional samples analysed or/and different quality of imputation. Additionally, in our study, we identified only one signal in the chr7q11.21 region, which in Lu et al.[17] was reported as two independent loci, *TMEM248* (also called as *C7orf42*) and *VKORC1L1*. Interestingly, the variant found in our study (rs34557764) lies in *RABGEF1* with established eQTL effect in 90 tissues, influencing the expression of various genes including *TMEM248* in testis, and *VKORC1L1* in skin, blood and esophagus muscularis[25,26], and further studies will be needed to ascertain the associated target gene(s). Overall, we report 44 loci harbouring 54 independent CCT-associated SNPs. These associations explain 8.5% and 7.2% of narrow sense CCT heritability in the European and Asian populations, respectively. Despite the small increase in the variance explained in the present study (~0.2%), the new loci greatly improved our understanding of potential underlying mechanisms.

At the newly identified CCT loci, if we select the nearest gene to the top SNP, we can putatively identify genes related to collagen and ECM (*ADAMTS2, ADAMTS8, COL6A2, COL12A1, FBN1, LOXL2, LUM/DCN/KERA, THSB2*), skeletal morphogenesis (*RUNX2*), embryonic development and cell growth (*FGF1*), TGF-β signalling (*TGFB2, LTBP1*), binding processes (*ARVCF, STAG*), coagulation and fibrinolysis systems (*HABP2*), endocytic machinery (*STON2*) and mitochondrial processes (*NDUFAF6*). It is important to stress that for several of these genes, the nearest gene may not be the relevant gene because the associated SNPs can have their primary effect on a more distant gene or genes. However, for a subset of the above nearest genes, additional information is available to support the noted gene. For example, knockout mouse models available for these genes have shown a variety of cornea-related phenotypes, including thin corneal stroma (*FBN1, KERA, LUM, TGFB2*)[27–32], corneal opacity (*LUM*)[30–32], absence of corneal endothelium (*TGFB2*)[27], delayed corneal endothelium maturation and increased thickness (*COL12A1*)[33]. While in other mouse models, observed phenotypes included fragile skin (*ADAMTS2, DCN, LUM*)[30,31,34,35] or bone abnormalities (*RUNX2, COL12A1*)[33,36]. Moreover,

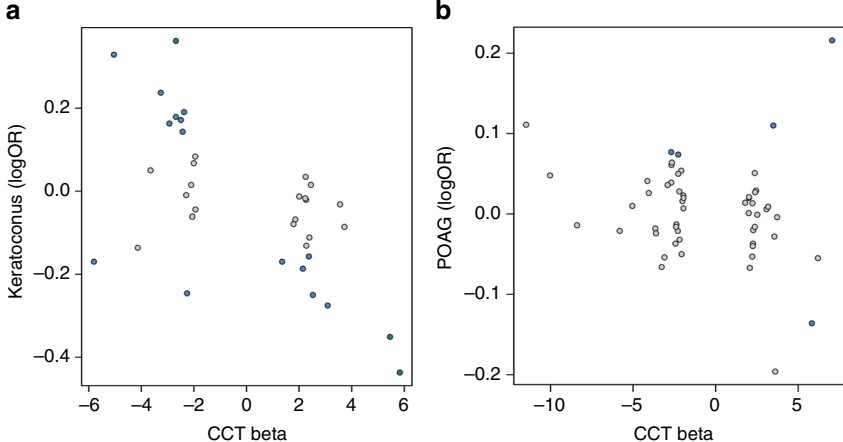

**Fig. 2** Correlation of effect sizes between CCT and keratoconus and CCT and primary-open angle glaucoma. Each dot represents a CCT-associated variant. In green, variants that surpassed the Bonferroni-corrected significance threshold ($P < 5.56 \times 10^{-4}$) in the keratoconus analysis (**a**). In blue, variants that were associated with a nominal level of significance ($P < 0.05$) in the keratoconus or primary-open angle glaucoma analysis. In gray, variants that did not show association with keratoconus or primary-open angle glaucoma ($P > 0.05$). **a** shows correlation of effect sizes between CCT and keratoconus, **b** shows correlation of effect sizes between CCT and primary-open angle glaucoma

connections between ECM, skeletal and TGF-β signalling pathways give support for the implicated genes to influence CCT. For instance, it has been reported that fibrillin (encoded by *FBN*1) plays an important role in the ECM by controlling TGF-β signalling[37]. In addition, the latent transforming growth factor β–binding protein 1 (encoded by *LTBP1*) is an ECM protein thought to mediate the binding between fibrillins and TGF-β, influencing the growth factors availability in bone and connective tissues.

Another locus of interest harbouring genes related to ECM and collagen is the "DCN" locus (top SNP rs7308752) in which *DCN*, *KERA* and *LUM* genes are located. Decorin (encoded by *DCN*, the closest gene to rs7308752) is a leucine-rich proteoglycan that promotes the formation of collagen fibers but also binds to the various isoforms of TGF-β and fibronectins[38]. Mutations in *DCN* have been identified in congenital stromal corneal dystrophy[35,39]; which produces characteristic corneal opacity and increased corneal thickness[35], making it an excellent candidate CCT gene. However, rs7308752 also shows a significant cis-eQTL effect ($P = 3.1 \times 10^{-6}$, in adipose tissue), modifying the expression of *KERA*, a keratan sulfate proteoglycan vital for maintaining corneal transparency. Mutations in *KERA* cause cornea plana-2 (OMIM:217300)[40,41], a recessive corneal disorder characterised by flattening of the normally convex corneal surface. Additionally, *LUM*, another gene in the region (30 kb apart from rs7308752), is a member of the small interstitial proteoglycan gene family and has been implicated in high myopia[32,42]. In the ocular tissue database[43], all three genes showed high expression levels in the cornea, with *KERA* showing the highest expression.

Our gene-based analysis identified association of the *CDO1* gene with CCT. The CDO1 protein, is a cysteine dioxygenase type 1, involved in various metabolic pathways. Expression studies in mouse found that *CDO1* is overexpressed in cornea compared with the lens; and based on its function, may play a role in protection against oxidative stress[44]. The top variant, rs34869, leading association of *CDO1*, is an established eQTL in transformed fibroblasts[26], modifying the expression levels of *CDO1* and it is encompassed within promoter histone marks and DNase I hypersensitive sites in at least 20 tissues.

Corneal thinning is one of the clinical features of keratoconus. We found in the keratoconus analysis a consistent direction of effect in 77% (28/36) of the CCT-associated SNPs (Fig. 2a). This finding suggests that the effect of variants on keratoconus is mediated

through their effect on CCT. We did not observe the same trend in the POAG analyses (Fig. 2b), with our data providing no support for a role for CCT SNPs in determining POAG risk.

Interestingly, besides the "DCN" locus, three other loci harbour genes implicated in Mendelian diseases including rare connective tissue, inflammatory and eye disorders with corneal thinning as one of their clinical features, giving weight for them to be prioritized in follow-up studies. The cross-ancestry GWAS revealed an intronic variant (rs8030753) in the *FBN1* gene. Mutations in *FBN1* are the major cause of Marfan syndrome, an autosomal dominant disorder characterized by multiple manifestations in the ocular, skeletal, and cardiovascular systems. Patients with Marfan syndrome have flattened corneas with reduced stromal thickness[45]. Common genetic variants in *FBN1* have also been associated with ocular refraction[46]. The rs8030753 shows a significant cis-eQTL effect modifying the expression of *FBN1* in whole blood ($P = 4.1 \times 10^{-10}$)[26]. Furthermore, we identified an intronic variant in *ADAMTS2*, which encodes a metalloproteinase involved in collagen metabolism[47]; Mutations in *ADAMTS2* have been found in patients with EDS, type VIIC[48], a recessive inherited connective-tissue disorder. We also identified a common CCT variant (rs4846476) in *TGFB2*. It has been shown that *TGFB2* is down-regulated in skin fibroblasts of brittle cornea syndrome patients carrying *PRDM5* mutations[49]. Our analysis brings the number of CCT-associated loci implicated in Mendelian diseases to nine, representing 20.5% (9/44) of the CCT loci. Most of the Mendelian disorders genes (8/9) are located within a 200 kb window from the lead SNP (Supplementary Table 12) with the exception of *AGBL*, located −784 kb away from rs4843040 in the 15q25.3 CCT-locus. Studies correlating gene variation to gene expression have found that most of the enhancers are located within a 200 kb window[50,51], supporting the hypothesis that lead CCT–associated SNPs might have an impact on the expression of genes that cause rare eye and connective tissue disorders. Our study reveals a considerable proportion of Mendelian genes as candidate genes involved in a quantitative trait.

Although findings of pathway analyses remain speculative, our exhaustive analyses suggest that the leading pathways implicated in CCT are related to the function and metabolism of connective tissue (e.g., collagen, ECM and basement membrane), as well as the regulation of TGF-β signalling, the development of skeletal system, and the ERAD pathway.

In conclusion, we have identified 19 novel loci associated with CCT and novel independent signals in six known loci. Together, CCT loci clearly point to genes implicated in collagen related pathways and ECM metabolism. The enrichment analyses highlighted gene-sets involved in collagen fibrils, ECM organisation, TGF-β signalling and fibroblastic determination, all fitting with a largely stromal contribution to CCT. Functional studies need to be performed to confirm which gene or genes are relevant at each locus and to assess the underlying mechanisms by which genetic variation influences CCT some of which promise to inform on the risk of complex diseases such as keratoconus.

## Methods

**Study design and sample description**. We performed meta-analyses of 1000 genomes phase 1 (integrated variant set- March 2012 release) imputed GWASs on CCT and tested significance of associations in keratoconus and POAG cohorts for lead CCT SNPs. The overall study design and main findings are depicted in Supplementary Fig. 1. In total, 19 CCT cohorts ($N = 25{,}910$) from the International Glaucoma Genetics consortium (IGGC) participated in this study. In stage 1, we performed a meta-analysis of cohorts of European ancestry (14 cohorts, $N = 17{,}803$). In stage 2, genome-wide significant variants ($P < 5 \times 10^{-8}$) from stage 1 were tested in a meta-analysis of cohorts of Asian ancestry (5 cohorts, $N = 8107$). We then performed in stage 3, a joint meta-analysis of European-specific and Asian-specific results. The individual cohorts are as described in detail in previous publications[18,52], with summary statistics and imputation details in Supplementary Table 1. To investigate the role of the identified CCT loci in keratoconus and POAG, we then tested the implicated CCT loci in disease case-control sets. The keratoconus datasets comprised cases and controls from Australia (711 keratoconus cases and 2622 controls from the Blue Mountains Eye Study) and the United States (222 keratoconus cases and 3324 controls). The POAG cases and controls were drawn from studies in Australia (1155 cases and 1992 controls) and the United States (3853 cases and 33480 controls). Detailed information of the keratoconus and POAG cohorts can be found in the Supplementary Note 1. The local research and medical ethics committees approved the individual studies. Written informed consent was obtained from all participants (or parents in case of minors) in accordance with the Declaration of Helsinki.

**Ancestry-specific and cross-ancestry GWAS meta-analyses**. All participating studies performed association testing under an additive model for the effect of the risk allele while adjusting for age, sex and at least the first five principle components for the population-based studies. In samples with related individuals association testing accounting for family structure was conducted using the –fastAssoc option in MERLIN[53] or the –mmscore[54] option implemented in GenABEL[55]. Before meta-analysis, we removed variants with MAF < 0.01, and with imputation quality scores less than 0.3.

Ancestry specific meta-analyses (European-specific and Asian-specific) and joint meta-analysis were performed using the inverse variance fixed effect scheme implemented in the software METAL[56]. The 'genomic inflation' correction option was used in METAL[56] and applied to all input files. We also computed the test statistics for heterogeneity of effect among studies for each variant using Cochran's Q-test. We removed variants with heterogeneity $P$-value < 0.001 from both European-specific and Asian-specific meta-analyses. Moreover, we focused on variants that were present in more than 25% of participating studies in the European-specific meta-analysis (at least four studies) and the Asian-specific meta-analysis (at least two studies). Finally, to detect additional loci associated with CCT, we performed a fixed-effect cross ancestry meta-analysis.

**Selecting independent variants**. We applied the conditional and joint (CoJo) analysis approach[57] implemented in the software Genome-wide Complex Trait Analysis[58] (GCTA) on European-specific meta-analysis results in order to identify potentially independent signals within the same genomic regions. For this CoJo analysis we used 1000 genomes phase 1 imputed data from Blue Mountain Eye Study (BMES) population cohort comprising 2582 individuals of European ancestry to calculate linkage disequilibrium (LD) patterns. We used the software GTOOL-v0.7.5 (http://www.well.ox.ac.uk/%7Ecfreeman/software/gwas/gtool.html) to convert BMES IMPUTE2 data (both SNPs and Indels) to the plink format. This conversion changes A/T/G/C/I/D/R based allele coding to 1 or 2 (first and second allele). We extracted variant IDs, hg19 genomic locations and converted A/T/G/C/I/D/R (from 1/2 based allele coding) for all 16,666,330 available variants (MAF > 0.01) from BMES data and merged it with the European-specific meta-analysis result file based on hg19 genomic location. Further quality checking was done by plotting the allele frequencies of the allele 1 of variants in chromosome 22 in the BMES cohort and European-specific meta-analysis summary file. The edited European-specific meta-analysis summary file with 1/2 allele coding was used as an input for the CoJo analysis. In the CoJo analysis we considered $5 \times 10^{-8}$ as the genome-wide significant threshold. We did not perform CoJo analysis in the Asian studies because the various Asian sub-studies (Indian, Malay, Chinese) had

differing ancestry within Asia and we did not have access to a suitably large (i.e., $N > 2000$) reference genotype data set for each Asian sub-population.

**Gene-based analysis**. We performed gene-based association testing using the VEGAS2 software[20]. VEGAS2 is an extension of the VErsatile Gene-based Association Study (VEGAS) approach[59] that uses 1000 genomes reference data to estimate LD between variants and provides a test using a more flexible gene boundary. For this analysis, we considered '−10 kbloc' parameter, which assigns all variants in the gene or within 10 kb on either side of a gene's transcription site to compute a gene-based $P$-value. We performed analysis using the default '-top 100' test that uses all (100%) variants assigned to a gene to compute gene-based $P$-value. We used 1000 Genomes phase 1 European and Asian populations to compute LD between variants for European-specific and Asian-specific gene-based analysis respectively. Finally, we meta-analysed the European-specific and Asian-specific gene-based results using Fisher's method for combining $P$-values.

**Analysis of case-control cohorts**. We tested the lead CCT-associated SNPs in two keratoconus datasets with 933 cases and 5946 controls and two POAG datasets with 5008 cases and 35,472 controls. Details of the disease cohorts can be found in the Supplementary Note 1. For both keratoconus and POAG we meta-analysed the association results for individual study samples using a fixed effect approach. The significance threshold for replication was established using the Bonferroni method for multiple testing correction.

**CCT-associated loci and Mendelian diseases**. We assessed whether the CCT-associated loci overlapped with candidate genes for rare Mendelian diseases. For this analysis we downloaded the most up-to-date annotations of genes to the Mendelian disease from the Online Mendelian Inheritance in Man (OMIM) portal[60] on 26th July 2016. We converted the genomic locations of CCT-associated variants from hg19 (or GRCh37) to the GRCh38 human genome build using the software liftOver[61,62] and extracted all gene transcription start sites that lie within the 1 mega-base (Mb) on either side of a given variant.

**Identifying regulatory variants**. Using the software HaploReg (version 4.1)[25] and RegulomeDB v1.1[22], we investigated regulatory annotations for variants in LD ($r^2 > 0.8$, 1000 genomes CEU) with the CCT-associated SNPs. To prioritize functional SNPs, we first used HaploRegv4.1 to extract all variants in LD with the 54 independent index SNPs and examined whether variants overlapped with regulatory elements of the ENCODE data, with the caveat that those do not include corneal tissue or cell lines data. We then used the RegulomeDB score to assess their potential functional consequence, as described previously[63]. Tissue-enrichment and gene prioritization analyses were performed with the DEPICT[23] framework, using independent CCT genome-wide significant SNPs. We also investigated the expression of functionally relevant genes in associated loci using the Ocular Tissue Database, https://genome.uiowa.edu/otdb/, in which gene expression is indicated as Affymetrix Probe Logarithmic Intensity Error (PLIER) normalized value (with normalization in PLIER as described in Wagner et al.[43]). Further, we used GARFIELD to assess enrichment of CCT association signals in regulatory features, using the 1005 features extracted from ENCODE, GENCODE and Roadmap Epigenomics projects provided by the software developers.

**Pathways analysis based on VEGAS2 gene-based $P$-values**. We adopted the resampling approach to perform pathway analyses using VEGAS2 derived gene-based $P$-values considering a '−10 kbloc' and '−200 kbloc' parameters respectively. The latter was performed to capture a larger number of nearby genes, in case the causal SNP or SNPs operate via long distance effects on genes in the wider region. The resampling approach performs a competitive test in which each gene-set is benchmarked against the 'typical' set of the same size. For individual gene-set, firstly we computed observed summed $\chi^2$ statistics by converting gene-based $P$-values of annotated genes to upper tail $\chi^2$ statistics with 1 degree of freedom. If two or more genes in a gene-set were located less than 500 kb of individual transcription sites, then only one gene was selected when computing the observed summed $\chi^2$ statistics and the other neighbouring genes were dropped out. This step might lead to loss of information but it ensures that the association of a gene-set is not driven by variants in LD. Following this, the same numbers of genes as present in a given gene set were repeatedly drawn at random from all set of genes used in the study and summed to generate the distribution of expected summed $\chi^2$ statistics. Finally, the empirical $P$-value of association of a gene-set is computed by comparing the observed summed $\chi^2$ statistics against the distribution of expected summed $\chi^2$ statistics using following formula:

$$\text{EmpP} = \frac{\sum_1^N I(\chi^{2*} \geq \chi^2) + 1}{N + 1}$$

where $I()$ is an indicator function which denotes whether a summed $\chi^2$ statistics from a random draw ($\chi^{2*}$) was equal to or more than the observed summed $\chi^2$ statistics ($\chi^2$), and $N$ is the total number of random draws performed to compute the distribution of expected summed $\chi^2$ statistics.

We performed separate European-specific and Asian-specific pathway analysis, and further combined the two ethnic-specific pathway $P$-values using the Fisher's method. For pathway analysis, we considered the Biosystem's pathways or gene-sets comprising minimum 10 and maximum 1000 genes[64]. In total 9981 gene-sets with 16,503 unique annotated genes were analysed. The processed Biosystems pathway/gene-set annotation file is available on the VEGAS2 webpage (https://vegas2.qimrberghofer.edu.au/).

**Pathway analysis using INRICH.** We also tested if single variants with association $P$-values less than $1 \times 10^{-4}$ were enriched in the Biosystem's pathways. For this analysis, we used the INRICH approach, which assumes a hypergeometric distribution for the null hypothesis that a pathway is not enriched with associated variants. To create LD-independent genomic regions to be tested for enrichment, we performed LD clumping with PLINK (--clump-p1 $1 \times 10^{-4}$ --clump-p2 0.05 --clump-r2 0.5 --clump-range-border 20), using the 1000 Genomes European and Asian reference data, for European-specific and Asian-specific gene-set enrichment analyses respectively.

**Pathway analysis using Ingenuity Pathway Analysis.** To select the genes included in the IPA®, we extracted the genes $+/- 200$ kb of the lead SNP and further chose those that were expressed in the cornea (Supplementary Table 12). Corneal expression levels were retrieved from the Ocular Tissue Database[43]. A gene list including 162 genes was used to run the IPA "Core-Analysis". Parameters of the analysis included (1) the Ingenuity Knowledge Base (Genes only) as the reference set, (2) including direct and indirect relationships, (3) experimentally observed, (4) from mammal species, (5) using all tissues and cell lines. Results were corrected for multiple testing using the Benjamini-Hochberg multiple testing correction as implemented in IPA.

**URLs.** GARFIELD software is available in a standalone version at http://www.ebi.ac.uk/birney-srv/GARFIELD/ and as a Bioconductor package at http://bioconductor.org/packages/release/bioc/html/garfield.html.

**Data availability.** Summary association statistics results that support the findings of this study have been deposited in http://hdl.handle.net/10283/2976.

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

## Acknowledgements

We gratefully thank the invaluable contributions of all study participants and staff at the recruitment centers. Complete funding information and acknowledgements for each individual study can be found in the Supplementary Note 2.

## Author contributions

Y.B., R.H., C.E.W., O.P., P.M., J.L.H., L.S.K., C.H., K.D.T., J.I.R., N.G.M., T.Z., R.A.M., S.E.S., S.EM.L., M.E.B., J.F.W., A.G.U., E.N.V., P.J.F., A.W.H., L.R.P., G.W.M., C.C.W.K., T.A., N.P., D.A.M., C.J.H., C.-Y.C., J.E.C., Y.S.R., J.L.W., K.P.B., C.MvD., and S.M., contributed samples. A.I.I., A.M.,V.V., R.H., H.S., G.C-P., P.G., J.N.C-B., X.L., S.Y., A.N., A.P.K., Y.C.T., Y.S., E.S., E.M.VL., P.B., I.S., T.B., S.EM.L., J.H.K., P.G.H., C.C.K., K.P.B and S.M., were involved in data analysis. D.S., L.R.P., C.M.vD., J.L.W., S.M., K.P.B., C.C.W.K., provided funding. Blue Mountains Eye Study- GWAS Group, NEIGHBORHOOD Consortium, and Wellcome Trust Case Control Consortium 2 (WTCCC2) were responsible for study-specific data. A.I.I., A.M.,V.V., L.R.P., D.A.M., J.L.W., C.MvD., and S.M., drafted the manuscript. Y.B., D.S., J.L.H., J.I.R., J.J., T.Y.W., A.W.H., L.R.P., T.A., D.A.M., C.J.H., C.-Y.C., J.E.C., Y.S.R., K.P.B., C.MvD., and S.M., conceived the study. All authors critically reviewed the manuscript.

## Additional information

**Competing interests:** The authors declare no competing interests.

Adriana I. Iglesias[1,2,3], Aniket Mishra[4], Veronique Vitart[5], Yelena Bykhovskaya[6,7], René Höhn[8,9], Henriët Springelkamp[1], Gabriel Cuellar-Partida[10], Puya Gharahkhani[10], Jessica N. Cooke Bailey[11,12], Colin E. Willoughby[13,14], Xiaohui Li[15,16], Seyhan Yazar[5,17], Abhishek Nag[18], Anthony P. Khawaja[19,20], Ozren Polašek[21], David Siscovick[22,23], Paul Mitchell[24], Yih Chung Tham[25], Jonathan L. Haines[11,12], Lisa S. Kearns[26], Caroline Hayward[5], Yuan Shi[25], Elisabeth M. van Leeuwen[1], Kent D. Taylor[15,16], Blue Mountains Eye Study—GWAS group, Pieter Bonnemaijer[1], Jerome I. Rotter[15,16], Nicholas G. Martin[27], Tanja Zeller[28,29], Richard A. Mills[30], Emmanuelle Souzeau[30], Sandra E. Staffieri[26], Jost B. Jonas[31], Irene Schmidtmann[32], Thibaud Boutin[5], Jae H. Kang[33], Sionne E.M. Lucas[34], Tien Yin Wong[25,35,36], Manfred E. Beutel[37], James F. Wilson[5,38], NEIGHBORHOOD Consortium, Wellcome Trust Case Control

Consortium 2 (WTCCC2), André G. Uitterlinden[2,39,40], Eranga N. Vithana[25], Paul J. Foster [20], Pirro G. Hysi [18], Alex W. Hewitt [26,41], Chiea Chuen Khor [42], Louis R. Pasquale [33,43], Grant W. Montgomery [27,44], Caroline C.W. Klaver [1,2,45], Tin Aung[25,35,36], Norbert Pfeiffer[8], David A. Mackey[17], Christopher J. Hammond [18], Ching-Yu Cheng[25,35,36], Jamie E. Craig[30], Yaron S. Rabinowitz[6,7], Janey L. Wiggs [43], Kathryn P. Burdon [34], Cornelia M. van Duijn[2] & Stuart MacGregor [10]

[1]Department of Ophthalmology, Erasmus Medical Center, 3000 CA, Rotterdam, The Netherlands. [2]Department of Epidemiology, Erasmus Medical Center, 3000 CA, Rotterdam, The Netherlands. [3]Department of Clinical Genetics, Erasmus Medical Center, 3000 CA, Rotterdam, The Netherlands. [4]University of Bordeaux, Bordeaux Population Health Research Center, INSERM UMR 1219, F-33000 Bordeaux, France. [5]Institute of Genetics and Molecular Medicine, Medical Research Council Human Genetics Unit, University of Edinburgh, EH42XU Edinburgh, UK. [6]Regenerative Medicine Institute and Department of Surgery, Cedars-Sinai Medical Center, CA 90048, Los Angeles, CA, USA. [7]Cornea Genetic Eye Institute, CA 90048, Los Angeles, CA, USA. [8]Department of Ophthalmology, University Medical Center Mainz, 55131 Mainz, Germany. [9]Department of Ophthalmology, Inselspital, University Hospital Bern, University of Bern, Bern CH-3010, Switzerland. [10]Statistical Genetics, QIMR Berghofer Medical Research Institute, QLD 4029, Brisbane, Australia. [11]Department of Population and Quantitative Health Sciences, Case Western Reserve University, OH 44106, Cleveland, OH, USA. [12]Institute for Computational Biology, Case Western Reserve University, Cleveland, OH 44106, USA. [13]Biomedical Sciences Research Institute, Ulster University, BT52 1SA Belfast, Northern Ireland, UK. [14]Royal Victoria Hospital, Belfast Health and Social Care Trust, BT12 6BA Belfast, Northern Ireland, UK. [15]Institute for Translational Genomics and Population Sciences and Department of Pediatrics, Los Angeles Biomedical Research Institute at Harbor-UCLA Medical Center, Torrance, CA 90509, CA, USA. [16]Division of Genomic Outcomes, Departments of Pediatrics and Medicine, Harbor-UCLA Medical Center, Torrance, CA 90502, CA, USA. [17]Centre for Ophthalmology and Visual Science, University of Western Australia, Lions Eye Institute, WA 6009, Perth, WA, Australia. [18]Department of Twin Research and Genetic Epidemiology, King's College London, WC2R 2LS London, UK. [19]Department of Public Health and Primary Care, Institute of Public Health, University of Cambridge School of Clinical Medicine, CB2 0SR Cambridge, UK. [20]NIHR Biomedical Research Centre, Moorfields Eye Hospital NHS Foundation Trust and UCL Institute of Ophthalmology, EC1V 9EL London, UK. [21]Faculty of Medicine, University of Split, HR-21000 Split, Croatia. [22]Departments of Medicine and Epidemiology and Cardiovascular Health Research Unit, University of Washington, WA 98101, Washington, USA. [23]The New York Academy of Medicine, NY 10029, New York, NY, USA. [24]Centre for Vision Research, Department of Ophthalmology and Westmead Institute for Medical Research, University of Sydney, NSW 2145, Sydney, NSW, Australia. [25]Singapore Eye Research Institute, Singapore National Eye Centre, 168751 Singapore, Singapore. [26]Centre for Eye Research Australia, University of Melbourne, Royal Victorian Eye and Ear Hospital, VIC 3002, East Melbourne, Australia. [27]Department of Genetics and Computational Biology, QIMR Berghofer Medical Research Institute, QLD 4029, Brisbane, Australia. [28]Department of General and Interventional Cardiology, University Heart Center Hamburg, 20251 Hamburg, Germany. [29]German Center for Cardiovascular Research (DZHK), Partner Site Hamburg/Kiel/Lübeck, 20246 Hamburg, Germany. [30]Department of Ophthalmology, Flinders University, SA 5042, Adelaide, Australia. [31]Department of Ophthalmology, Medical Faculty Mannheim of the Ruprecht-Karls-University of Heidelberg, 68167 Mannheim, Germany. [32]Institute for Medical Biostatistics, Epidemiology and Informatics, University Medical Center Mainz, 55131 Mainz, Germany. [33]Channing Division of Network Medicine, Brigham and Women's Hospital, Boston, MA 02115, MA, USA. [34]Menzies Institute for Medical Research, University of Tasmania, Hobart, TAS 7005, TAS, Australia. [35]Ophthalmology & Visual Sciences Academic Clinical Program (Eye ACP), Duke-NUS Medical School, 169857 Singapore, Singapore. [36]Department of Ophthalmology, Yong Loo Lin School of Medicine, National University of Singapore, Singapore 117549, Singapore. [37]Department of Psychosomatic Medicine and Psychotherapy, University Medical Center Mainz, Mainz 55131, Germany. [38]Centre for Global Health Research, Usher Institute for Population Health Sciences and Informatics, University of Edinburgh, EH16 4UX Edinburgh, UK. [39]Department of Internal Medicine, Erasmus Medical Center, 3000 CA, Rotterdam, The Netherlands. [40]Netherlands Consortium for Healthy Ageing, Netherlands Genomics Initiative, 2593 HW, The Hague, The Netherlands. [41]School of Medicine, Menzies Institute for Medical Research, University of Tasmania, Hobart, TAS 7005, TAS, Australia. [42]Genome Institute of Singapore, 60 Biopolis Street, Singapore 138672, Singapore. [43]Department of Ophthalmology, Harvard Medical School and Massachusetts Eye and Ear Infirmary, Boston, MA 02114, MA, USA. [44]Institute for Molecular Bioscience, University of Queensland, QLD 4067, Brisbane, Australia. [45]Department of Ophthalmology, Radboud University Medical Center, 6525 GA, Nijmegen, The Netherlands. These authors contributed equally: Adriana I. Iglesias, Aniket Mishra,Veronique Vitart. These authors jointly supervised this work: Jamie E. Craig, Yaron S. Rabinowitz, Janey L. Wiggs, Kathryn P. Burdon, Cornelia M. van Duijn, Stuart MacGregor.

## Blue Mountains Eye Study—GWAS group

Jie Jin Wang[24], Elena Rochtchina[24], John Attia[46], Rodney Scott[46], Elizabeth G. Holliday[46], Tien Yin Wong[26], Paul N. Baird[26], Jing Xie[26], Michael Inouye[47], Ananth Viswanathan[20] & Xueling Sim[36]

[46]University of Newcastle, Newcastle, NSW 2308, Australia. [47]Walter and Eliza Hall Institute of Medical Research, Melbourne, VIC 3052, VIC, Australia.

## NEIGHBORHOOD Consortium

R. Rand Allingham[48], Murray H. Brilliant[49], Donald L. Budenz[50], William G. Christen[51], John Fingert[52,53], David S. Friedman[54], Douglas Gaasterland[55], Terry Gaasterland[56], Michael A. Hauser[48,57], Peter Kraft[58],

Richard K. Lee[59], Paul R. Lichter[60], Yutao Liu[48,57], Stephanie J. Loomis[43], Sayoko E. Moroi[60], Margaret A. Pericak-Vance[61], Anthony Realini[62], Julia E. Richards[60], Joel S. Schuman[63], William K. Scott[61], Kuldev Singh[64], Arthur J. Sit[65], Douglas Vollrath[66], Robert N. Weinreb[67], Gadi Wollstein[63], Donald J. Zack[54] & Kang Zhang[67]

[48]Department of Ophthalmology, Duke University Medical Center, Durham, NC, 27705 NC, USA. [49]Center for Human Genetics, Marshfield Clinic Research Foundation, Marshfield, WI, USA. [50]Department of Ophthalmology, University of North Carolina, Chapel Hill, NC, 27517 NC, USA. [51]Department of Medicine, Brigham and Women's Hospital, Boston, MA 02114 MA, USA. [52]Department of Ophthalmology, College of Medicine, University of Iowa, Iowa City, IA, 52242 IA, USA. [53]Department of Anatomy/Cell Biology, College of Medicine, University of Iowa, Iowa City, IA, 52242 IA, USA. [54]Wilmer Eye Institute, John Hopkins University, Baltimore, MD, 21287 MD, USA. [55]Eye Doctors of Washington, Chevy Chase, MD, 20815 MD, USA. [56]Scripps Genome Center, University of California at San Diego, San Diego, CA, 92037 CA, USA. [57]Department of Medicine, Duke University Medical Center, Durham, NC, 27710 NC, USA. [58]Department of Biostatistics, Harvard School of Public Health, Boston, MA, 02115 MA, USA. [59]Bascom Palmer Eye Institute, University of Miami Miller School of Medicine, Miami, FL, 33136 FL, USA. [60]Department of Ophthalmology and Visual Sciences, University of Michigan, Ann Arbor, MI, 48105 MI, USA. [61]Institute for Human Genomics, University of Miami Miller School of Medicine, Miami, FL, 33136 FL, USA. [62]Department of Ophthalmology, WVU Eye Institute, Morgantown, WV, 26506 WV, USA. [63]Department of Ophthalmology, UPMC Eye Center, University of Pittsburgh, Pittsburgh, PA, 15213 PA, USA. [64]Department of Ophthalmology, Stanford University, Palo Alto, CA, 94303 CA, USA. [65]Department of Ophthalmology, Mayo Clinic, Rochester, MN, 55905 MN, USA. [66]Department of Genetics, Stanford University, Palo Alto, CA, 94305 CA, USA. [67]Department of Ophthalmology, Hamilton Eye Center, University of California, San Diego, CA, 92093 CA, USA

## Wellcome Trust Case Control Consortium 2 (WTCCC2)

Peter Donnelly[68,69], Ines Barroso[70], Jenefer M. Blackwell[71,72], Elvira Bramon[73], Matthew A. Brown[74], Juan P. Casas[75,76], Aiden Corvin[77], Panos Deloukas[70], Audrey Duncanson[78], Janusz Jankowski[79,80,81], Hugh S. Markus[82], Christopher G. Mathew[83], Colin N.A. Palmer[84], Robert Plomin[85], Anna Rautanen[68], Stephen J. Sawcer[86], Richard C. Trembath[83], Nicholas W. Wood[87], Chris C.A. Spencer[68], Gavin Band[68], Céline Bellenguez[68], Colin Freeman[68], Garrett Hellenthal[68], Eleni Giannoulatou[68], Matti Pirinen[68], Richard Pearson[68], Amy Strange[68], Zhan Su[68], Damjan Vukcevic[68], Cordelia Langford[70], Sarah E. Hunt[70], Sarah Edkins[70], Rhian Gwilliam[70], Hannah Blackburn[70], Suzannah J. Bumpstead[70], Serge Dronov[70], Matthew Gillman[70], Emma Gray[70], Naomi Hammond[70], Alagurevathi Jayakumar[70], Owen T. McCann[70], Jennifer Liddle[70], Simon C. Potter[70], Radhi Ravindrarajah[70], Michelle Ricketts[70], Matthew Waller[70], Paul Weston[70], Sara Widaa[70] & Pamela Whittaker[70]

[68]Wellcome Trust Centre for Human Genetics, University of Oxford, Roosevelt Drive, Oxford OX37BN, UK. [69]Dept Statistics, University of Oxford, Oxford OX1 3TG, UK. [70]Wellcome Trust Sanger Institute, Wellcome Trust Genome Campus, Hinxton, Cambridge CB10 1SA, UK. [71]Telethon Institute for Child Health Research, Centre for Child Health Research, University of Western Australia, 100 Roberts Road, Subiaco, WA 6008, Australia. [72]Cambridge Institute for Medical Research, University of Cambridge School of Clinical Medicine, Cambridge CB2 0XY, UK. [73]Department of Psychosis Studies, NIHR Biomedical Research Centre for Mental Health at the Institute of Psychiatry, King's College London and The South London and Maudsley NHS Foundation Trust, Denmark Hill, London SE5 8AF, UK. [74]University of Queensland Diamantina Institute, Brisbane, QLD 4102, QLD, Australia. [75]Dept Epidemiology and Population Health, London School of Hygiene and Tropical Medicine, London WC1E 7HT, UK. [76]Dept Epidemiology and Public Health, University College London, London WC1E 6BT, UK. [77]Neuropsychiatric Genetics Research Group, Institute of Molecular Medicine, Trinity College Dublin, Dublin 2, Eire, Ireland. [78]Molecular and Physiological Sciences, The Wellcome Trust, London NW1 2BE, UK. [79]Department of Oncology, Old Road Campus, University of Oxford, Oxford OX3 7DQ, UK. [80]Digestive Diseases Centre, Leicester Royal Infirmary, Leicester LE7 7HH, UK. [81]Centre for Digestive Diseases, Queen Mary University of London, London, E1 2AD, UK. [82]Clinical Neurosciences, St George's University of London, London SW17 0RE, UK. [83]King's College London Dept Medical and Molecular Genetics, King's Health Partners, Guy's Hospital, London SE1 9RT, UK. [84]Biomedical Research Centre, Ninewells Hospital and Medical School, Dundee DD1 9SY, UK. [85]King's College London Social, Genetic and Developmental Psychiatry Centre, Institute of Psychiatry, Denmark Hill, London SE5 8AF, UK. [86]University of Cambridge Dept Clinical Neurosciences, Addenbrooke's Hospital, Cambridge CB2 0QQ, UK. [87]Dept Molecular Neuroscience, Institute of Neurology, Queen Square, London WC1N 3BG, UK.

