## [Peer Review File · Nature Communications]

Reviewer #1 (Remarks to the Author):

The manuscript by Iglesias and colleagues describes a meta-analysis of GWAS studies for CCT. The study describes analysis of imputed genotypes from 14 studies comprising 17,803 individuals of European ancestry (stage 1), that identified 28 loci for CCT that reached genome wide significance. Seven of these 28 loci have not been previously implicated, and the closest genes are reported. The 28 lead SNPs from stage 1 were then analysed in an Asian specific ancestry meta-analysis (8,107 individuals), none reached genome-wide significance, nevertheless, 19 SNPs were nominally significant with the same direction for CCT influence as the European cohort. Next, they performed a cross-ancestry fixed-effect meta-analysis to detect additional loci (stage 3). Nineteen novel loci identified in this stage of the analysis included 4 of the novel loci from stage 1. Interestingly, one of the novel loci identified in stage 1 and 3 (European-specific) was LTBP1.

To identify additional variants, the authors performed analyses for loci with multiple independent variants, and gene-based (autosomal) association, that revealed significant association of CDO1 with CCT in Europeans.

In the next phase of the study, the authors explored the influence of CCT-associated variants with risk of keratoconus (34 SNPs in 933 cases and 5,946 controls) and POAG (54 SNPs in 5,008 cases and 35,472 controls). For keratoconus, three were significant with the expected direction of effect and have been previously reported, and for POAG, none were significantly associated.

Next, the authors determined that 9/44 CCT associated loci lie within 1 Mb of rare Mendelian disease genes implicated in corneal or connective tissue disorders. The 54 SNPs and an additional 920 SNPs in LD were prioritised for evaluation of their location within regulatory regions, and tissue expression enrichment. Gene-set enrichment highlighted 7 of the closest genes to the 44 CCT loci implicated from the cross-ethnic meta-analysis are amongst the 64 fibroblastic signature genes, and protein interaction networks highlighted direct and indirect interactions. Connections between ECM, skeletal and TGF-beta signalling pathways support the genes implicated in CCT.

The manuscript is well written with a logical flow, and reports robust data that identifies new loci and implicated genes that influence CCT. The multi-analytical approach described highlights the links between implicated loci and genes known to cause rare connective tissue and corneal Mendelian disorders, that synergises current knowledge with the analyses presented here. Importantly, this study also highlights the lack of correlation between CCT and POAG.

Suggested changes

The abstract would benefit from some clarification, for example use of PAOG rather than glaucoma, and specify that significant correlation with keratoconus was for 3 previously reported loci.

Page 7, line 188, low-frequency variants, this should be defined in the main text

Page 10, line 266-267, 118 were prioritised, an explanation of how these were prioritised in the main text would be helpful here

Page 11, line 277-281, an expanded description of the tissue enrichment would be beneficial, e.g. were specific cell types in these tissues sub-categorized

Page 12 line 319, typographical error, change no to not

Reviewer #2 (Remarks to the Author):

Iglesias et al. presented a genetic study for identifying and understanding genetic variants of central corneal thickness (CCT). They performed genome-wide association study on European ancestry and Asian ancestry cohorts separately and then performed meta-analysis to combine results. They conducted comprehensive analyses including single SNP test, gene-based test, conditional analysis and functional characterization. They identified 19 novel associated regions that have not been reported before for CCT. In addition, they performed association analysis on lead SNPs from CCT study in keratoconus cohorts and POAG cohorts and show a significant correlation in effect size between CCT and keratoconus but not between CCT and glaucoma. The manuscript is well written and the study is reasonably designed. I have the following comments and suggestions:

1. With increased sample size, it is not too surprising that more CCT-associated SNPs will be discovered and they have small effect sizes. It is a good addition to existing findings of CCT. Although it may be beyond the scope of this paper, functional experiments on any of the novel genes for understanding the underlying mechanism of CCT is lacking in this study. Authors did a lot of database search for functional characterization. Can author add a bit more discussion and literature search if some novel genes are involved in functional experiments that are relevant to CCT in animal model?
2. In the abstract, it says "Using CCT associated index variants, we found a significant correlation in effect sizes between CCT and keratoconus ($r = -0.61$, $P = 5.30 \times 10^{-5}$) but not between CCT and glaucoma ($r = -0.17$, $P = 0.2$). This supports a causal relationship for the former pair but not the latter". I don't think that the evidence supports a causal relationship, but just showing that CCT and keratoconus may be more relevant phenotypically or have high correlation. CCT is continuous and keratoconus is binary, so the effect size is not comparable and correlation may be affected by large values of effect size. Is keratoconus or glaucoma status also available in some CCT studies? If so, it would be interesting to see how CCT is correlated with keratoconus and glaucoma (e.g. boxplots of CCT in cases and controls).
3. It is worthwhile to perform a gene-level pathway analysis on significant or top genes from meta-analysis (e.g. DAVID and/or IPA) in addition to protein-protein interaction analysis. It is also interesting to perform the analysis (e.g. on top 100 genes) separately in Asian and European cohorts and check if there are overlapping pathways.
3. What is the distribution of CCT, is it normal distributed? Are they measured similarly across different studies and countries? Was any transformation performed before association analysis? It looks like the range of CCT are similar across all studies except TwinsUK study, which has a much lower left bound (211).
4. In lines 438 and 439, could the author elaborate on how the family structure was handled in the analysis for the studies that have family data? And how sample size or standard errors for family-based study is specified in METAL?
5. If a variant is filtered (e.g. low frequency or imputation quality) in one study, would this study be excluded from meta-analysis? If so, the sample size for each variant will be different. It would be good to list the actual sample size for each variant in table 1.
6. The imputation is based on 1000 Genomes Project phase 1 data. Phase 3 has been released for a while, is there any benefit if using the updated panel? Some discussion is needed.
7. A general minor problem about scientific notation. Could the author delete the 0s (e.g., 10-08 to 10-8)?
8. In line 173, the author may still need to adjust for multiple testing. In other words, 0.05/28

instead of 0.05.

9. A typo in line 737, $P < 5.56$.

10. In the Manhattan plots, can the author add the gene names, which could be easier for readers to see the gene locations?

Reviewer #3 (Remarks to the Author):

This study reports GWAS meta-analysis for central corneal thickness (CCT). This is a very well designed and executed large-scale GWAS study, which included over 25,000 individuals of both European and Asian ancestries. The study used several statistical genetics approaches to scan for associations between genetic variation and CCT, as well as to analyze associations between CCT, POAG, and keratoconus. The authors describe mapping of 19 new loci for CCT and an interesting link between CCT and keratoconus. The finding of a genetic association between genes that control CCT and keratoconus is of particular importance since etiology of keratoconus is poorly understood. Overall the study is well done and the paper is very well written. The enthusiasm for this manuscript is high. There are several issues listed below, however, that need to be addressed.

1. The location of the strongest SNP in close proximity to a gene is not sufficient to declare that gene a candidate. Even though this is a common practice, a more scientifically prudent approach would be just to say something like "a gene nearest to the SNP" and list all genes 200 kb upstream and 200 kb downstream of the SNP in a supplementary table. Literature suggests that the majority of enhancers are located within 200 kb of genes.

2. Analysis of association with genes underlying Mendelian disorders is a bit of an overstretch. Even though some enhancers can be located as far as 1 Mb from a gene, the majority of enhancers would be within 200 kb of a gene. Therefore, 400-kb critical region would be more appropriate for such analysis.

9. DAPPLE data analyzing protein-protein interactions between "candidate gene" is the weakest part of the manuscript. You really do not have enough information to perform gene network or protein network analysis. As mentioned above, the proximity of a gene to a SNP is not a proof that the gene is affected/involved. It is a pure speculation to base network analysis on such flimsy grounds. If this analysis is left in the manuscript, it should be moved to the supplement (including Fig. 3).

10. I would suggest to rearrange the figures. I suggest to move Fig. 1 and 3 to the supplement. Move Manhattan plots to the main manuscript. Add graph showing lack of correlation between CCT and POAG to Fig. 2.

7. DEPICT tissue data are not very useful. They do not add any valuable information and I believe they should be removed.

8. The term gene-set enrichment analysis is usually used in a different context. What you are analyzing here is an enrichment for the gene ontology terms. Please, rewrite this section to make this clear.

1. The figures with Manhattan plots need proper legends. Please, clearly define the significance thresholds shown in the figures and provide corresponding clarification in the text of the manuscript.

4. Please, provide figure legends for all supplementary figures.
3. There is an error in the Bonferroni-corrected P-value in the Figure 2 legend.
6. Supplementary Figure 6 needs a more detailed legend.

Response to Referees:

NCOMMS-17-14943 (Iglesias, Mishra, Vitart et al)

We would like to express our thanks to the reviewers for their constructive comments. We have modified the text and the figures based on reviewer's comments. Our specific responses to reviewer questions are described below

Reviewer #1

Remarks to the Author

The manuscript by Iglesias and colleagues describes a meta-analysis of GWAS studies for CCT. The study describes analysis of imputed genotypes from 14 studies comprising 17,803 individuals of European ancestry (stage 1), that identified 28 loci for CCT that reached genome wide significance. Seven of these 28 loci have not been previously implicated, and the closest genes are reported. The 28 lead SNPs from stage 1 were then analysed in an Asian specific ancestry meta-analysis (8,107 individuals), none reached genome-wide significance, nevertheless, 19 SNPs were nominally significant with the same direction for CCT influence as the European cohort. Next, they performed a cross-ancestry fixed-effect meta-analysis to detect additional loci (stage 3). Nineteen novel loci identified in this stage of the analysis included 4 of the novel loci from stage 1. Interestingly, one of the novel loci identified in stage 1 and 3 (European-specific) was LTBP1.

To identify additional variants, the authors performed analyses for loci with multiple independent variants, and gene-based (autosomal) association, that revealed significant association of CDO1 with CCT in Europeans.

In the next phase of the study, the authors explored the influence of CCT-associated variants with risk of keratoconus (34 SNPs in 933 cases and 5,946 controls) and POAG (54 SNPs in 5,008 cases and 35,472 controls). For keratoconus, three were significant with the expected direction of effect and have been previously reported, and for POAG, none were significantly associated.

Next, the authors determined that 9/44 CCT associated loci lie within 1 Mb of rare Mendelian disease genes implicated in corneal or connective tissue disorders. The 54 SNPs and an additional 920 SNPs in LD were prioritised for evaluation of their location within regulatory regions, and tissue expression enrichment. Gene-set enrichment highlighted 7 of the closest genes to the 44 CCT loci implicated from the cross-ethnic meta-analysis are amongst the 64 fibroblastic signature genes, and protein interaction networks highlighted direct and indirect interactions. Connections between ECM, skeletal and TGF-beta signalling pathways support the genes implicated in CCT.

The manuscript is well written with a logical flow, and reports robust data that identifies new loci and implicated genes that influence CCT. The multi-analytical approach described highlights the links between implicated loci and genes known to cause rare connective tissue and corneal Mendelian disorders, that synergises current knowledge with the analyses presented here. Importantly, this study also highlights the lack of correlation between CCT and POAG.

Suggested changes

Question #1

The abstract would benefit from some clarification, for example use of PAOG rather than glaucoma, and specify that significant correlation with keratoconus was for 3 previously reported loci.

Response: In the revised version of the abstract, we have clarified that we referred to primary open-angle glaucoma.

The significant correlation in effect sizes between CCT and keratoconus is not driven only by the previously reported keratoconus loci. To demonstrate our hypothesis, we have removed 4 known keratoconus SNPs (rs35193497 close to *ZNF469*, rs2755238 close to *FOXO1*, rs3132303 close to *COL5A1* and rs66720556 between *MPDZ* and *NFIB*) from the plot and re-calculated the correlation (see Figure below, panel B). The SNP in *FNDC3B* was not available in the keratoconus look-up. After removing the 4 known keratoconus SNPs, the correlation coefficient does not change dramatically (i.e. from $r = -0.621$, $P = 5.30 \times 10^{-5}$ to $r = -0.611$, $P = 2.04 \times 10^{-4}$). Therefore, we have not specified that significant correlation is only due to previously reported keratoconus SNPs. We have added a sentence in the results section noting that the correlation is essentially unchanged if the known SNPs in *ZNF469*, *FOXO1*, *COL5A1* and *MPDZ/NFIB* are removed.

Results section, page 8-9

“Overall, we found a significant negative correlation of effect sizes across CCT and keratoconus ($r = -0.62$, $P = 5.30 \times 10^{-5}$) (Figure 2A, Supplementary Table 5), this correlation was largely unchanged if the known SNPs in *ZNF469*, *FOXO1*, *COL5A1* and *MPDZ/NFIB* were removed from the analysis ($r = -0.61$, $P = 2.04 \times 10^{-4}$)”

A)

B)

Question #2

Page 7, line 188, low-frequency variants, this should be defined in the main text

Response: We have changed the text accordingly.

Results section, page 7

“Two of the 44 loci are driven by low-frequency variants (i.e. $0.01 < \text{minor allele frequency [MAF]} < 0.05$)”

Question #3

Page 10, line 266-267, 118 were prioritised, an explanation of how these were prioritised in the main text would be helpful here

Response: We have described how the 118 SNPs were prioritized

Results section, page 10

“Of these, 118 were prioritized including the 54 lead SNPs and another 64 SNPs which were selected based on their RegulomeDB score²² (i.e., 1a-1f, 2a-2c or 3a). SNPs with a score from 1a-1f to 2a-2c were classified as showing maximum evidence for being located in regulatory regions, while SNPs with a score of 3a were classified as showing medium evidence (Supplementary Table 8)”

Question #4

Page 11, line 277-281, an expanded description of the tissue enrichment would be beneficial, e.g. were specific cell types in these tissues sub-categorized

Response: We have provided an extended description of the tissue enrichment results and specified the system, tissue and cell types identified.

Results section, page 11

“...Further, we tested if genes in associated CCT loci were highly expressed in any of the 209 Medical Subject Heading (MeSH) annotations used in DEPICT²⁴. Tissue-enrichment analyses showed 33 FDR-associated (<0.05) tissues or cell type annotations. Of these, one annotation included the musculoskeletal system, five included tissues such as muscle and connective tissue, and nine included cell types such as myocytes, osteoblast, chondrocytes, mesenchymal stem cells, stromal cells and fibroblasts (Supplementary Table 9)”

Question #5

Page 12 line 319, typographical error, change no to not

Response: We have corrected this typographical error (page 13)

Reviewer #2

Remarks to the Author:

Iglesias et al. presented a genetic study for identifying and understanding genetic variants of central corneal thickness (CCT). They performed genome-wide association study on European ancestry and Asian ancestry cohorts separately and then performed meta-analysis to combine results. They conducted comprehensive analyses including single SNP test, gene-based test, conditional analysis and functional characterization. They identified 19 novel associated regions that have not been reported before for CCT. In addition, they performed association analysis on lead SNPs from CCT study in keratoconus cohorts and POAG cohorts and show a significant correlation in effect size between CCT and keratoconus but not between CCT and glaucoma. The manuscript is well written and the study is reasonably designed. I have the following comments and suggestions:

Question #1

With increased sample size, it is not too surprising that more CCT-associated SNPs will be discovered and they have small effect sizes. It is a good addition to existing findings of CCT. Although it may be beyond the scope of this paper, functional experiments on any of the novel genes for understanding the underlying mechanism of CCT is lacking in this study. Authors did a lot of database search for functional characterization. Can author add a bit more discussion and literature search if some novel genes are involved in functional experiments that are relevant to CCT in animal model?

Response: We appreciate the comment from the reviewer, we agree that functional studies provide additional information regarding biological mechanisms, although there is a limit to the amount of material we can include in this single paper. Nonetheless, we have incorporated the reviewer suggestion and performed a literature search for the novel genes which is now included in the discussion.

Discussion section, page 14

“...Knockout mouse models available for these genes have shown a variety of cornea-related phenotypes, including thin corneal stroma (FBNI, KERA, LUM, TGFB2)²⁹⁻³⁴, corneal opacity (LUM)³²⁻³⁴, absence of corneal endothelium (TGFB2)²⁹, delayed corneal endothelium maturation and increased thickness (COL12A1)³⁵. While in other mouse models, observed phenotypes included fragile skin (ADAMTS2, DCN, LUM)^{32,33,36,37} or bone abnormalities (RUNX2, COL12A1)^{35,38}”

Question #2

In the abstract, it says “Using CCT associated index variants, we found a significant correlation in effect sizes between CCT and keratoconus ($r = -0.61$, $P = 5.30 \times 10^{-05}$) but not between CCT and glaucoma ($r = -0.17$, $P = 0.2$). This supports a causal relationship for the former pair but not the latter”. I don't think that the evidence supports a causal relationship, but just showing that CCT and keratoconus may be more relevant phenotypically or have high correlation. CCT is continuous and keratoconus is binary, so the effect size is not comparable and correlation may be affected by large values of effect size. Is keratoconus or glaucoma status

also available in some CCT studies? If so, it would be interesting to see how CCT is correlated with keratoconus and glaucoma (e.g. boxplots of CCT in cases and controls).

Response: Our original thinking in terms of causality was in terms of the methods outlined in Pickrell et al Nat Genet, 48, 709-717, 2016, where genome-wide significant hits for each of trait 1 and trait 2 are used to attempt to infer the likely causal relation between trait 1 and 2. In practice here, we have many genome-wide significant SNPs for CCT but not for keratoconus. Our result where genome-wide CCT SNPs tend to also affect keratoconus suggests a causal relationship but we cannot rule out either keratoconus causing CCT changes (in a subset of individuals) or there being another unobserved trait that causally influences both CCT and keratoconus. In any event, due to these difficulties in establishing causality, we have removed the claims regarding causality in the abstract and in the discussion.

Keratoconus or glaucoma status is not available to us in the CCT studies to enable us to draw the suggested boxplots

We have removed the following sentence from the abstract, page 4

“This supports a causal relationship for the former pair but not the latter.”

We have also changed the discussion section, page 15

From

“This finding suggests that the effect of variants on keratoconus is mediated through their effect on CCT and provides supporting evidence that CCT is related to keratoconus in a causal manner”

To

“This finding suggests that the effect of variants on keratoconus is mediated through their effect on CCT”

Question #3

It is worthwhile to perform a gene-level pathway analysis on significant or top genes from meta-analysis (e.g. DAVID and/or IPA) in addition to protein-protein interaction analysis. It is also interesting to perform the analysis (e.g. on top 100 genes) separately in Asian and European cohorts and check if there are overlapping pathways.

Response: We already conducted substantial work on pathway analysis approaches in the paper and hence feel that adding more would be unlikely to make major improvements to the paper. To our knowledge, a flaw in DAVID is the outdated annotations – these have been reported to cause issues in interpretation of findings – see <http://www.biorxiv.org/content/early/2016/04/19/049288> . An issue with the IPA analysis is that it assumes that all genes in the pathway are independent – this assumption can be incorrect because in some pathways the genes are physically close to each other and are hence not independent (are correlated due to linkage disequilibrium).

Question #4

What is the distribution of CCT, is it normal distributed? Are they measured similarly across different studies and countries? Was any transformation performed before association analysis? It looks like the range of CCT are similar across all studies except TwinsUK study, which has a much lower left bound (211).

Response: CCT has a distribution which is close to normal and no transformation was applied prior to analysis.

We have checked the distribution of CCT in the TwinsUK study (see histogram below). The mean CCT was 546.47 (SD = 33.46), min value = 430.3 (not 211) and max value = 657.5. These values are comparable with the distribution found in all other cohorts, we have corrected this number in the supplementary data and apologize for the misunderstanding.

Question #5

In lines 438 and 439, could the author elaborate on how the family structure was handled in the analysis for the studies that have family data? And how sample size or standard errors for family-based study is specified in METAL?

Response: In the revised version of the manuscript, we have explained in the methods section how family structure was handled.

Methods section, page 19

“In samples with related individuals association testing accounting for family structure was conducted using the `-fastAssoc` option in MERLIN⁴⁵ or the `-mmscore`⁴⁶ option implemented in GenABEL⁴⁷”

Question #6

If a variant is filtered (e.g. low frequency or imputation quality) in one study, would this study be excluded from meta-analysis? If so, the sample size for each variant will be different. It would be good to list the actual sample size for each variant in table 1.

Response: Indeed, if a variant is filtered out in one study, the study is excluded from the meta-analyses. As suggested by the reviewer we have added the sample size for each variant in Table 1.

Question #7

The imputation is based on 1000 Genomes Project phase 1 data. Phase 3 has been released for a while, is there any benefit if using the updated panel? Some discussion is needed.

Whilst an updated reference set would potentially improve results, at this time we would not be able to re-impute all cohorts to more recent reference set, which would reduce our sample size. In the future all consortium members will impute to updated reference panels but this would form a separate publication. We note that the Haplotype Reference Consortium reference set is predominantly aimed at European ancestry populations and would be largely irrelevant to the Asian ancestry populations we study here.

Question # 8

A general minor problem about scientific notation. Could the author delete the 0s (e.g., 10-08 to 10-8)?

Response: We have deleted the 0s in all the scientific notations.

Question #9

In line 173, the author may still need to adjust for multiple testing. In other words, 0.05/28 instead of 0.05

Response: We have corrected for multiple test and adjust the text accordingly

Results section, page 6

“In stage 2, we examined the 28 lead SNPs from stage 1 in the Asian-specific meta-analysis ($n = 8,107$) and found that 16, including the novel lead SNPs within or close to ADAMTS8 and DCN, were significant after Bonferroni correction ($P \leq 1.79 \times 10^{-3}$, $0.05/28$), further 3 other SNPs including the two novel close to STAG1 and NDUF6 were nominally significant ($P < 0.05$). The effect estimates of these 19 (16+3) loci were in the same direction and order of magnitude as in the European-specific meta-analysis (Table 1 and Supplementary Table 2)”

Question #10

A typo in line 737, $P < 5.56$.

Response: We have corrected the text.

Figure 2, page 29

“...In green, variants that surpassed the Bonferroni-corrected significance threshold ($P < 5.56 \times 10^{-4}$) in the keratoconus analysis”

Question #11

In the Manhattan plots, can the author add the gene names, which could be easier for readers to see the gene locations?

Response: We have added the gene names to all three Manhattan plots. The cross-ancestry Manhattan plot is now included as main **Figure 1**

Reviewer #3

Remarks to the Author:

This study reports GWAS meta-analysis for central corneal thickness (CCT). This is a very well designed and executed large-scale GWAS study, which included over 25,000 individuals of both European and Asian ancestries. The study used several statistical genetics approaches to scan for associations between genetic variation and CCT, as well as to analyze associations between CCT, POAG, and keratoconus. The authors describe mapping of 19 new loci for CCT and an interesting link between CCT and keratoconus. The finding of a genetic association between genes that control CCT and keratoconus is of particular importance since etiology of keratoconus is poorly understood. Overall the study is well done and the paper is very well written. The enthusiasm for this manuscript is high. There are several issues listed below, however, that need to be addressed.

Question #1

The location of the strongest SNP in close proximity to a gene is not sufficient to declare that gene a candidate. Even though this is a common practice, a more scientifically prudent approach would be just to say something like “a gene nearest to the SNP” and list all genes 200 kb upstream and 200 kb downstream of the SNP in a supplementary table. Literature suggests that the majority of enhancers are located within 200 kb of genes.

Response: We fully agree with the comment from reviewer#3. Close proximity to a gene is not sufficient to declare that gene as a candidate. Therefore, we have added in the footnotes of Table 1, Table 2 and Supplementary Tables 2, 3, 5 and 6, the following sentence: *“Nearest gene (reference NCBI build37) is given as locus label, but this should not be interpreted as providing support that the nearest gene is the best candidate, a list including all the genes +/- 200kb of the lead SNP is presented in Supplementary Table 14”*

As suggested for the reviewer, we now provide **Supplementary Table 14**, which contains a list of all genes 200 kb upstream and 200 kb downstream of the lead SNP corresponding to Table 1. This supplementary table also lists the genes considered in the DAPPLE analysis (definition of the genes included in the DAPPLE analysis is explained in response to point 3 of reviewer#3)

Question #2

Analysis of association with genes underlying Mendelian disorders is a bit of an overstretch. Even though some enhancers can be located as far as 1 Mb from a gene, the majority of enhancers would be within 200 kb of a gene. Therefore, 400-kb critical region would be more appropriate for such analysis.

Response: Mentioned analysis does not claim to explain the causal mechanism behind the associated variant, but simply report the list of genes underlying Mendelian diseases that are located within the CCT-associated loci, for which we used 1Mb as an arbitrary boundary. Given that the 1Mb arbitrary boundary provides a comprehensive list of genes, the information gathered from this analysis is highly informative for functional characterisation in future.

As pointed out by the reviewer cis-regulatory elements have been located as far as 1Mb from their target gene, hence this choice of 1Mb as arbitrary boundary provides a more comprehensive listing.

In our analysis we found that nine of the CCT loci are within 1Mb of genes implicated in rare corneal or connective tissue disorders. In fact, 88.8% (8/9) of the genes implicated in Mendelian disorders found in our analysis were +/- 200 kb from the signal (see table below), the exception being *AGBL1* - 784kb away from the *AKAP13* CCT locus. This should be viewed as hypothesis generating rather than claim of any functional connection; hence, we argue that using a larger critical region casts a broader net.

lead SNP	Chr:bp	Mendelian disease causing gene	Mendelian disorder	Distance from lead SNP
rs96067	1:36571920	COL8A2	Fuchs endothelial dystrophy	6.1kb 5' of COL8A2
rs4846476	1:218526228	TGFB2	Loeys-Dietz syndrome	intronic
rs35028368	5:178671146	ADAMTS2	Ehlers-Danlos syndrome, type VIIC	intronic
rs3132303	9:137444298	COL5A1	Ehlers-Danlos syndrome, classic type	89kb 5' of COL5A1
rs7308752	12:91527181	DCN-KERA	DCN(corneal dystrophy) KERA (Cornea plana 2)	12kb 3' of DCN and 75kb 5' of KERA
rs8030753	15:48801935	FBNI	Marfan syndrome	intronic
rs12912010	15:67467143	SMAD3	Loeys-Dietz syndrome	intronic
rs4843040	15:85838636	AGBL1	Fuchs endothelial dystrophy	784kb from AGBL1
rs35193497	16:88324821	ZNF469	Brittle cornea syndrome	169kb 5' of ZNF469

We have discussed this finding in the discussion section and included the table above as **Supplementary Table 13**

Discussion section, page 16

*“Our analysis brings the number of CCT-associated loci implicated in Mendelian diseases to nine, representing 20.5% (9/44) of the CCT loci. Most of the Mendelian disorders genes (8/9) are located within a 200 kb window from the lead SNP (Supplementary Table 13) with the exception of *AGBL1*, located -784kb away from the *AKAP13* CCT locus. Studies correlating gene variation to gene expression have found that most of the enhancers are located within a 200 kb window^{52,53}, supporting the hypothesis that lead CCT-associated SNPs might have an impact on the expression of genes that cause rare eye and connective tissue disorders”*

Question #3

DAPPLE data analyzing protein-protein interactions between “candidate gene” is the weakest part of the manuscript. You really do not have enough information to perform gene network or

protein network analysis. As mentioned above, the proximity of a gene to a SNP is not a proof that the gene is affected/involved. It is a pure speculation to base network analysis on such flimsy grounds. If this analysis is left in the manuscript, it should be moved to the supplement (including Fig. 3).

Response: We disagree that the protein-protein interaction evidence is the weakest part of the manuscript and rather would argue that this is an uncommon and interesting feature for a complex trait analysis. We feel that we did not explain clearly what the analysis performed consisted of and hope that the edits will clarify this point.

The DAPPLE software looks for significant physical connectivity among proteins encoded by genes in loci associated to a trait according to protein-protein interactions reported in the literature.

The first sentence in the method description is ambiguous and the interactions are not drawn from our data, rather they are curated validated protein-protein interactions reported in the literature.

We have change the text in the methods section, page 23

From

“We used DAPPLE²⁵, which tests whether protein-protein interaction networks built from implicated regions are more connected than chance expectation”

To

“We used DAPPLE²⁵ to test for an enrichment in protein-protein interactions across loci, amongst the proteins encoded by the genes at the CCT loci, compared to chance expectation”.

As this enrichment is statistically significant, we feel that it is a result that should be kept in the manuscript. We have left the figure in the main text although if the journal editor believes that space is an issue we are happy for this to be moved to the Supplementary section.

A list of genes, not the closest gene to the index SNP, served as input for each locus. As described in the methods, these are “genes within the LD intervals defined by SNPs from the 1000 genome reference data in $r^2 > 0.5$ with the lead SNPs”. For some loci there are no genes defined using this criterion, hence only 29 out of the 44 loci were used in the test and, for some of the loci with genes, the proteins encoded are not in the curated list of protein-protein interactions, hence are also not used in the test. We now provide **Supplementary Table 14** for the genes encompassed at each genome-wide significant meta-analysis locus that were used as input in the DAPPLE analysis which should help clarify the process undertaken.

Question #4

I would suggest to rearrange the figures. I suggest to move Fig. 1 and 3 to the supplement. Move Manhattan plots to the main manuscript. Add graph showing lack of correlation between CCT and POAG to Fig. 2.

Response: As suggested by the reviewer, we have moved Figure 1 to the supplements, added the correlation between CCT and POAG to Figure 2 but kept Figure 3 in the main text (see our answer to question #3, reviewer 3). Further, we have moved the Manhattan plot of the cross-ancestry meta-analysis to the main text (now **Figure 1**) as this is the main result described and discussed. Manhattan plots of the Eur-only and ASN-only were kept in the Supplementary data.

Question #5

DEPICT tissue data are not very useful. They do not add any valuable information and I believe they should be removed.

Response: We have provided a more detailed description of the tissue enrichment results from DEPICT (see answer to question # 4, reviewer 1). Tissue enrichment analysis adds extra information supporting that genes in CCT-associated loci are expressed in relevant tissues and cells, including connective tissue, mesenchymal stem cells, stromal cells and fibroblasts. We have left this analysis in the manuscript although if considered by the journal editor we could move it to supplementary data.

Question #6

The term gene-set enrichment analysis is usually used in a different context. What you are analyzing here is an enrichment for the gene ontology terms. Please, rewrite this section to make this clear.

Response: We agree with the reviewer, the term “gene-set enrichment analysis” might confuse readers. We have change the term to “pathway analysis”. Therefore, we have adjusted the text accordingly in the methods, results and supplementary tables

Question #7

The figures with Manhattan plots need proper legends. Please, clearly define the significance thresholds shown in the figures and provide corresponding clarification in the text of the manuscript.

Response: We have provided legends to all QQ and Manhattan plots, in addition we have annotated each locus to the nearest gene as in Table 1 as requested by reviewer#2

Question #8

Please, provide figure legends for all supplementary figures.

Response: We have provided legends to all QQ and Manhattan plots.

Question #9

There is an error in the Bonferroni-corrected P-value in the Figure 2 legend.

Response: We have corrected the text accordingly (see question # 10 from reviewer 2)

Question #10

Supplementary Figure 6 needs a more detailed legend.

Response: We have clarified the legend of Supplementary Figure 6

Reviewer #1 (Remarks to the Author):

The authors have carefully and satisfactorily addressed all comments raised

Reviewer #2 (Remarks to the Author):

The authors have addressed most of my comments. There are still many places with typos and unfinished sentences. Authors need to check every sentence carefully. I listed a few below.

1. Abstract, “. . . Our findings reveal” has an extra period.
2. Page 15, “All three genes showed high expression levels in the cornea with, KERA showing the highest expression according to an ocular tissue database.” Is not complete.
3. Page 15, “CDO1 is a cysteine dioxygenase type 1, involved ...” missed “gene”?
4. Page 16, “Patients with Marfan syndrome patients have flattened corneas with reduced stromal thicknes” should remove “patients”.
5. Page 22, “further combined the two ethnic specific pathway”.
6. Page 23, Data availability. <https://datashare.is.ed.ac.uk/URLs> will URLs be available soon?

Reviewer #3 (Remarks to the Author):

1. I am still enthusiastic about the main findings of the manuscript, i.e., identification of novel loci for CCT and their association with keratoconus. However, my recommendations regarding pathway and, especially, protein network analyses have not been addressed. As I explained in my initial review, the proximity of a gene to a lead SNP is not sufficient grounds to assume causality. Several studies presented at the ASHG 2017 meeting explored the distance between lead SNPs and causal genes and found that that majority of causal genes are within 200 kb of the SNP, and are sometimes as far as 500 kb from the SNP. Therefore, all pathway and protein network data presented in the manuscript are very speculative. One way to address this issue would be to perform pathway and protein network analyses using genes within 200 kb (or even 500 kb) window of the SNP after filtering them for expression in relevant tissues (such as cornea, connective tissues etc), for example, or other functional metric. This would still be rather speculative, but at least would have strong rationale. Supplementary Table 14 is a good starting point for such analysis. Using purely statistical methods (such as VEGAS2 or LD) for identifying candidate genes for the pathway and protein network analysis is hardly convincing.

2. The authors still continue to refer to the nearest genes as causal genes in the text of the manuscript and, especially, in the discussion. In some cases, it seems to be justified by functional data from the literature, but in other cases is again based on proximity alone.

3. Please, rewrite the abstract in past tense.

4. Occasional grammatical errors are still present in the manuscript.

5. Please, verify supplementary figure numbers for consistency with the manuscript.

REVIEWERS' COMMENTS:

Reviewer #3 (Remarks to the Author):

All my concerns were properly addressed. No further comments.